# Structural basis for C-degron selectivity across KLHDCX family E3 ubiquitin ligases

Daniel C. Scott[1,6], Sagar Chittori [1,6], Nicholas Purser [2], Moeko T. King [1], Samuel A. Maiwald [3], Kelly Churion[1], Amanda Nourse [1], Chan Lee [4], Joao A. Paulo [4], Darcie J. Miller[1], Stephen J. Elledge [5], J. Wade Harper [4], Gary Kleiger [2,3] & Brenda A. Schulman [1,3] ✉

Specificity of the ubiquitin-proteasome system depends on E3 ligase-substrate interactions. Many such pairings depend on E3 ligases binding to peptide-like sequences - termed N- or C-degrons - at the termini of substrates. However, our knowledge of structural features distinguishing closely related C-degron substrate-E3 pairings is limited. Here, by systematically comparing ubiquity-lation activities towards a suite of common model substrates, and defining interactions by biochemistry, crystallography, and cryo-EM, we reveal principles of C-degron recognition across the KLHDCX family of Cullin-RING ligases (CRLs). First, a motif common across these E3 ligases anchors a substrate's C-terminus. However, distinct locations of this C-terminus anchor motif in different blades of the KLHDC2, KLHDC3, and KLHDC10 β-propellers establishes distinct relative positioning and molecular environments for substrate C-termini. Second, our structural data show KLHDC3 has a pre-formed pocket establishing preference for an Arg or Gln preceding a C-terminal Gly, whereas conformational malleability contributes to KLHDC10's recognition of varying features adjacent to substrate C-termini. Finally, additional non-consensus interactions, mediated by C-degron binding grooves and/or by distal propeller surfaces and substrate globular domains, can substantially impact substrate binding and ubiquitylatability. Overall, the data reveal combinatorial mechanisms determining specificity and plasticity of substrate recognition by KLDCX-family C-degron E3 ligases.

N- and C-degron pathways - wherein E3 ligases recognize specific motifs at protein termini - have emerged as major mediators of biological regulation[1–3]. Protein quality control depends on E3 ligases recognizing N- or C-terminal sequences aberrantly exposed in mistranslated, mis-processed, or mis-assembled proteins[4–7]. Furthermore, some signaling pathways rely on proteolytic cleavage events triggering degradation of one or both protein halves via recognition of their neo N- and/or C-terminus by N- and/or C-degron E3 ligases[5,8,9]. Moreover, due to their ligandability, the E3 ligases that recognize protein termini are also of great interest for targeted protein degradation[10–12].

Recently a number of cullin-RING ligase (CRL) E3s were discovered to recognize and mediate ubiquitin-mediated proteolysis of

[1]Department of Structural Biology, St. Jude Children's Research Hospital, Memphis, TN, USA. [2]Department of Chemistry and Biochemistry, University of Nevada, Las Vegas, Las Vegas, NV, USA. [3]Department of Molecular Machines and Signaling, Max Planck Institute of Biochemistry, Martinsried, Germany. [4]Department of Cell Biology, Harvard Medical School, Boston, MA, USA. [5]Division of Genetics, Brigham and Women's Hospital, Howard Hughes Medical Institute, Department of Genetics, Harvard Medical School, Boston, MA, USA. [6]These authors contributed equally: Daniel C. Scott, Sagar Chittori. ✉e-mail: schulman@biochem.mpg.de

proteins harboring specific N- and C-terminal degron motifs[7,13]. CRLs are multisubunit complexes[14]. The cullin subunit adopts an elongated structure that serves as a central scaffold. On one side, the cullin protein's N-terminal domain binds to an interchangeable substrate receptor (SR) module with distinct substrate-binding properties. For instance, a subset of BCbox-ELONGIN B-ELONGIN C (hereafter BCbox-EloB/C) complexes are SR modules that bind interchangeably to CUL2 to form CRL2^BCbox protein E3 ligase complexes, while another subset of BCbox-EloB/C complexes with CUL5 are termed CRL5^BCbox protein complexes[15-18]. The opposite end of the cullin associates with a RING domain-containing subunit (RBX1 for CUL2, and RBX2 for CUL5)[19,20]. A CRL's E3 ligase activity is stimulated by NEDD8 modification of the C-terminal domain of the cullin subunit[21,22]. Neddylation enables the CRL to partner with one of several potential ubiquitin-carrying enzymes (UBE2D-, UBE2R-, and one UBE2G-family E2s, or the E2 UBE2L3 together with an ARIH-family RBR E3) that either directly ubiquitylate the SR-bound substrate in reactions referred to as substrate priming, or extend a poly-ubiquitin chain from a substrate-linked ubiquitin[23-30].

The largest cohort of CRL SRs recognizing terminal degrons are the four BC-box proteins in the KLHDC-family (hereafter collectively referred to as KLHDCX), KLHDC1, KLHDC2, KLHDC3, and KLHDC10[31,32]. These SRs all feature Kelch-type 6-bladed β-propellers that recognize C-degrons bearing terminal Gly residues. Selectivity differences across the family have been attributed to the target protein's penultimate residue: a C-terminal Gly-Gly sequence is the consensus degron motif for CRL5^KLHDC1 and CRL2^KLHDC2; sequences terminating in Arg/Lys/Gln-Gly and Trp/Ala/Pro-Gly are consensus degron motifs recognized by the CRL2-based SRs KLHDC3 and KLHDC10, respectively[33].

The only structures showing details of KLHDCX family C-degron recognition are for KLHDC2[34]. A trio of KLHDC2 side-chains, Arg-Ser-Arg (RSR), in blade 4 specifically bind to the substrate's C-terminal di-Gly motif. Meanwhile, the so-called "N-chamber" of the KLHDC2 Kelch domain embraces a further 4−5 residues upstream of the di-Gly sequence. These latter residues are secured by backbone contacts, explaining why KLHDC2 can interact with substrates with diverse sequences upstream of the C-terminal di-Gly sequence[34].

Notably, KLHDC2 also accommodates C-terminal sequences subtly differing from the consensus, especially when other interactions also secure the binding partner[35-37]. This was demonstrated for a tetrameric KLHDC2-EloB/C self-assembly stabilized by the C-terminus of one KLHDC2 protomer, which terminates in Gly-Ser, occupying the degron-binding site in the adjacent protomer and additional intersubunit interactions involving KLHDC2 and EloC elements[35]. It also seems likely that additional contacts beyond the C-terminal sequence contribute to KLHDC2's substrate-binding domain's discrimination between NEDD8, which binds with high-affinity, and ubiquitin, whose binding is negligible despite nearly 60% sequence identity between NEDD8 and ubiquitin[35].

While structures showing details of other KLHDCX E3s binding their partner proteins have yet to be reported, several studies suggested an ability to bind diverse C-terminal sequences[13,33]. Global protein stability (GPS) screens, and recent identification of a new substrate[38], showed KLHDC3-dependent degradation activity conferred by distinct C-terminal degrons varying in their penultimate residues. Meanwhile, the ability of KLHDC10 to accommodate diverse sequences was revealed by discovery of its involvement in ribosome quality control pathways[39]. Proteins that stall on ribosomes during translation are recognized by the nuclear export mediator factor to facilitate recruitment of Ala-charged tRNA to extend nascent polypeptides with poly Ala tracts. These Ala tail proteins are subject to proteasomal degradation in part through C-terminal recognition by KLHDC10. Thus, KLHDC10 can also recognize non-consensus C-degrons[3,39,40].

While C-degron binding is a prerequisite for substrate recognition by KLHDCX family E3s[32,40-42], additional elements may impact substrate ubiquitylation efficiency. We had previously discovered that NEDD8, but not ubiquitin, avidly binds to the KLHDC2 substrate-binding pocket[35]. Furthermore, NEDD8 is ubiquitylated by an activated KLHDC2 mutant, although slow association with the E3 prevents its ubiquitylation by the wild-type KLHDC2[35]. Nonetheless, the differences in ubiquitin versus NEDD8 ubiquitylation (albeit by a KLHDC2 mutant) in these assays suggested that a systematic comparison of KLHDCX family E3 activity towards a common suite of UBLs could yield insights into substrate targeting.

Here we initially extend analyses of UBL substrate ubiquitylation efficiency across the KLHDCX cohort of E3s. Unexpectedly robust activity reveals distinct specificities across this E3 family, and enable structures of KLHDC3 and KLHDC10 bound to proteins harboring C-terminal glycines. Taken together with prior studies of KLHDC2 complexes, the structures reveal that KLHDCX family E3s utilize a common C-terminus anchor (sequence R/F-S-R), but achieve distinct recognition through display of this motif across spatially differing Kelch-repeat blades in their propeller domains. Thus, the data suggest rules of C-degron recognition across a family of structurally related E3s.

## Results

### Similar but distinct KLHDCX ubiquitylation profiles towards model protein substrates

We sought to systematically compare the efficiencies of KLHDCX-dependent substrate ubiquitylation in the presence of ubiquitin or ubiquitin-like proteins (UBLs; Fig. 1a), all containing C-terminal di-Gly or Gly amino acid sequence motifs (Fig. 1a). Previous studies have shown that optimal substrate ubiquitylation is achieved by matching substrate, SR, cullin and ubiquitin-carrying enzymes[1,23,24]. Although the rules for substrate recognition are emerging, this can depend not only on SR affinity for a degron but in some cases interactions with additional substrate features. Meanwhile, ubiquitylation also relies on pairing with a ubiquitin-carrying enzyme and can be influenced by features of the substrate including lysine accessibility[26,43-47], its SR, and/or the identity of the cullin-RBX complex[29,30]. For example, at one extreme, ARIH-family E3s efficiently prime a plethora of structurally diverse substrates with ubiquitin[43]. In contrast, priming of CRL1 substrates occurs slowly with UBE2R2[48]. Instead, UBE2R2 rapidly extends poly-ubiquitin chains onto ubiquitins already linked to substrates of many CRLs, and is also particularly efficient at modifying substrates of many CRL2 E3s[27,28,49,50]. As such, ubiquitin-carrying enzymes spanning the spectrum of known catalytic preferences were assayed with KLHDCX family members. Monomeric versions of KLHDCX proteins were employed to relieve potential autoinhibition without affecting ubiquitylation activity[35].

All of the KLHDCX family members promoted ubiquitylation of one or more of the UBL substrates tested. However, both target selectivity as well as the efficiency of ubiquitylation substantially differed despite the protein substrates sharing a common Gly-Gly C-terminal sequence (Fig. 1b−e, Supplementary Figs. 1a−d, compare UB, NEDD8, SUMO1, SUMO2, ISG15, FAT10, and URM1). Neddylated CUL2^KLHDC2 was the most selective CRL profiled, promoting ubiquitylation of the UBLs NEDD8, FAT10, and to a lesser extent URM1 (Fig. 1b, Supplementary Fig. 1a). Neddylated CUL5^KLHDC1 showed a similar reactivity profile, with the notable exceptions that UB was weakly targeted, and a lack of activity towards URM1 (Fig. 1e, Supplementary Fig. 1d). Interestingly, neddylated CUL2^KLHDC3 robustly ubiquitylated NEDD8 and UB substrates (Fig. 1c, Supplemenetary Fig. 1b) despite their C-degrons lacking a penultimate Arg or Gln residue (that previously had been identified as a determinant of the KLHDC3 consensus degron). Furthermore, in reactions with the ubiquitin-carrying

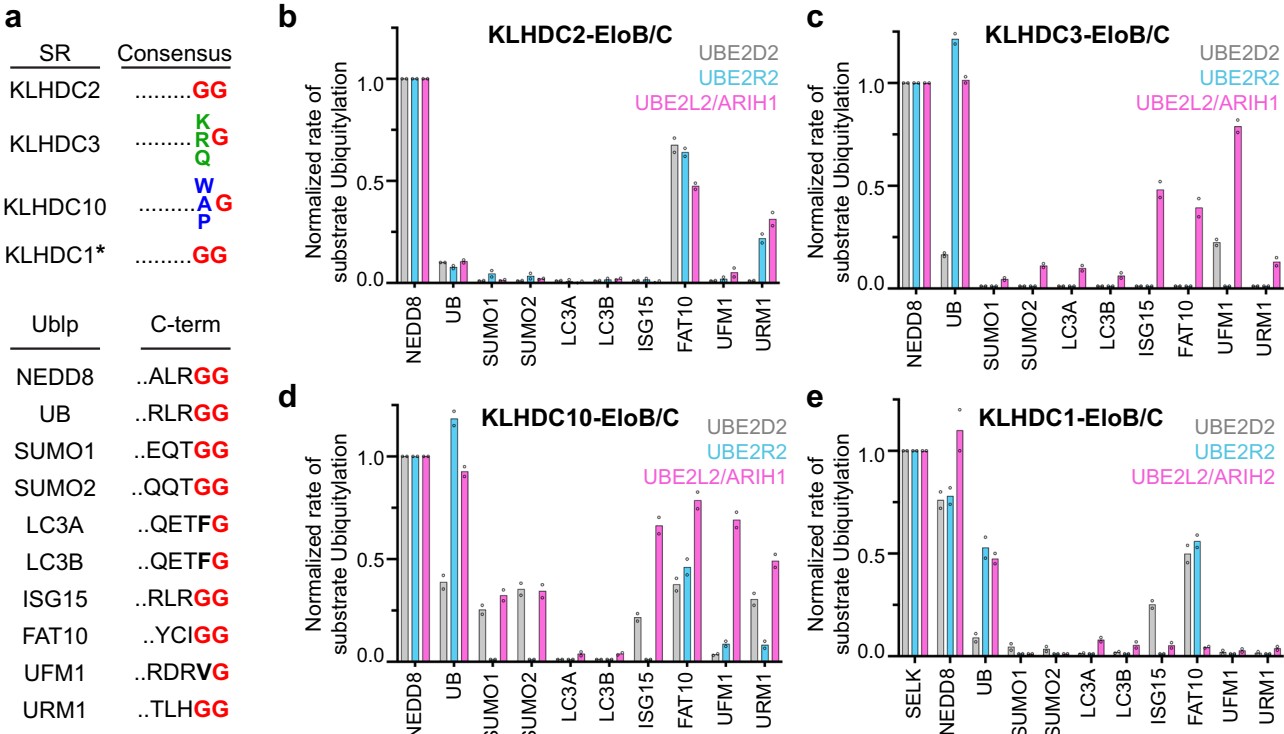

**Fig. 1 | KLHDCX family members mediate ubiquitylation of an assortment of Gly terminating substrates. a** C-terminal consensus degron sequences recognized by KLHDCX substrate receptor family members (top panel). *The consensus sequence of KLHDC1 has not been reported, but has been show to recognize degrons terminating in GlyGly. C-terminal sequences of UBLs used in this study (bottom panel). **b** Quantification of pulse-chase ubiquitylation assays monitoring monomeric CRL2^KLHDC2 dependent ubiquitylation of the indicated Gly terminating substrates by the ubiquitin-carrying enzymes UBE2D2 (gray), UBE2R2 (cyan), and UBE2L2/ARIH1 (purple). **c** Same as (**b**) but with monomeric CRL2^KLHDC3. **d** Same as (**b**) but with CRL2^KLHDC10. **e** Same as (**b**) but with CRL5^KLHDC1, and with the CRL5 specific UCE UBE2L2/ARIH2. Bar graphs are the average of $n = 2$ independent experiments. Source data are provided as a Source Data file.

enzymes UBE2L2/ARIH1, neddylated CUL2^KLHDC3 also supported ubiquitylation of ISG15, FAT10, UFM1, and URM1. While neddylated CUL2^KLHDC10 activity was most robust towards NEDD8 and UB, some activity was also observed towards all the UBLs except LC3A and LC3B that were not substantially modified by any KLHDCX E3s (Fig. 1d, Supplementary Fig. 1C).

In summary, KLHDCX family members display a remarkable degree of plasticity towards the ubiquitylation of model protein substrates with C-terminal Gly residues, hinting that substrate recognition may be influenced by features beyond the extreme C-terminus.

## Unique C-terminal Gly recognition by KLHDC3

To understand the basis for C-degron recognition by KLHDC3, we crystallized and solved the structure of a monomeric version of KLHDC3-EloB/C bound to ubiquitin at 2.0 Å resolution (Fig. 2a, Supplementary Fig. 2a, b, Supplementary Table 1). As expected, the substrate-binding domain of KLHDC3 forms a six-bladed β-propeller (Fig. 2b)[51,52]. Typically, each propeller blade is composed of four antiparallel β-strands, labeled A to D from the center to the outer rim of the propeller (Fig. 2b)[51]. Blades 1–5 contain β-strands that are contiguous in sequence, whereas blade 6 contains three strands that progress into the BC-box, and an outer so-called "Velcro" strand from the N-terminus of the domain that fastens the β-propeller. Together, the blades adopt a funnel-like propeller shape that engages three major surfaces of ubiquitin: the C-terminal di-Gly motif occupies the bottom of the funnel; ubiquitin's preceding C-terminal tail residues bind in the funnel stem; and ubiquitin's globular domain engages the top (Fig. 2c). The volume of the KLHDC3 substrate binding pocket appears to restrict the number of C-terminal residues that may be accommodated to approximately six amino acids. This latter feature is similar to other SR interactions with terminal degron motifs[34].

The KLHDC3 C-degron-binding site is formed by two distinct chambers, which are separated by a ridge-like feature and analogous to those in KLHDC2 termed the N- and C-chamber (Fig. 2d)[34]. Residues that precede the substrate's di-Gly bind in KLHDC3's N-chamber, while the degron's C-terminal di-Gly motif is buried deep within the C-chamber. At the bottom of the C-chamber, a basic patch engages the substrate's extreme C-terminal carboxyl group (Fig. 2e). Key contacts are made by a KLHDC3 RSR motif (Arg240, Ser241, and Arg292), all located within blade 5 (Fig. 2f, g). Mutations in KLHDC3's RSR motif severely impaired ubiquitylation, consistent with a critical functional role of the motif in substrate recognition (Fig. 2h, Supplementary Fig. 2c).

The structure explains KLHDC3's strict requirement for its substrates to terminate with a C-terminal Gly: its Cα is tightly wedged in the stem of the KLHDC3 funnel-like structure, which would exclude any side-chain larger than a hydrogen (Fig. 2g). Indeed, KLHDC3-mediated ubiquitylation is severely impaired upon replacing the substrate's C-terminal Gly with Ala or Asp (Fig. 2h, Supplementary Fig. 2d). The C-terminal di-Gly motif is further supported by KLHDC3 residues Asp16 and Arg293, located underneath the substrate at the bottom of the C-chamber, as well as Phe182, Val115, and Thr309 that flank both sides of the substrate (Fig. 2f, g). The penultimate Gly is stabilized by a hydrogen bond between its backbone carbonyl and Arg292 from KLHDC3's RSR motif.

Numerous KLHDC3 side-chains contact the C-degron backbone within the stem portion of KLHDC3 (Fig. 2f, g). Asn16 at the bottom of KLHDC3's N-chamber pocket forms a hydrogen bond with the carbonyl of ubiquitin's Arg74, and also interacts with the carbonyl of ubiquitin's Leu73 through a water-mediated hydrogen bond. KLHDC3 residue Arg198 forms a water-mediated hydrogen bond to the carbonyl of ubiquitin's Arg72. Electrostatic interactions are also observed

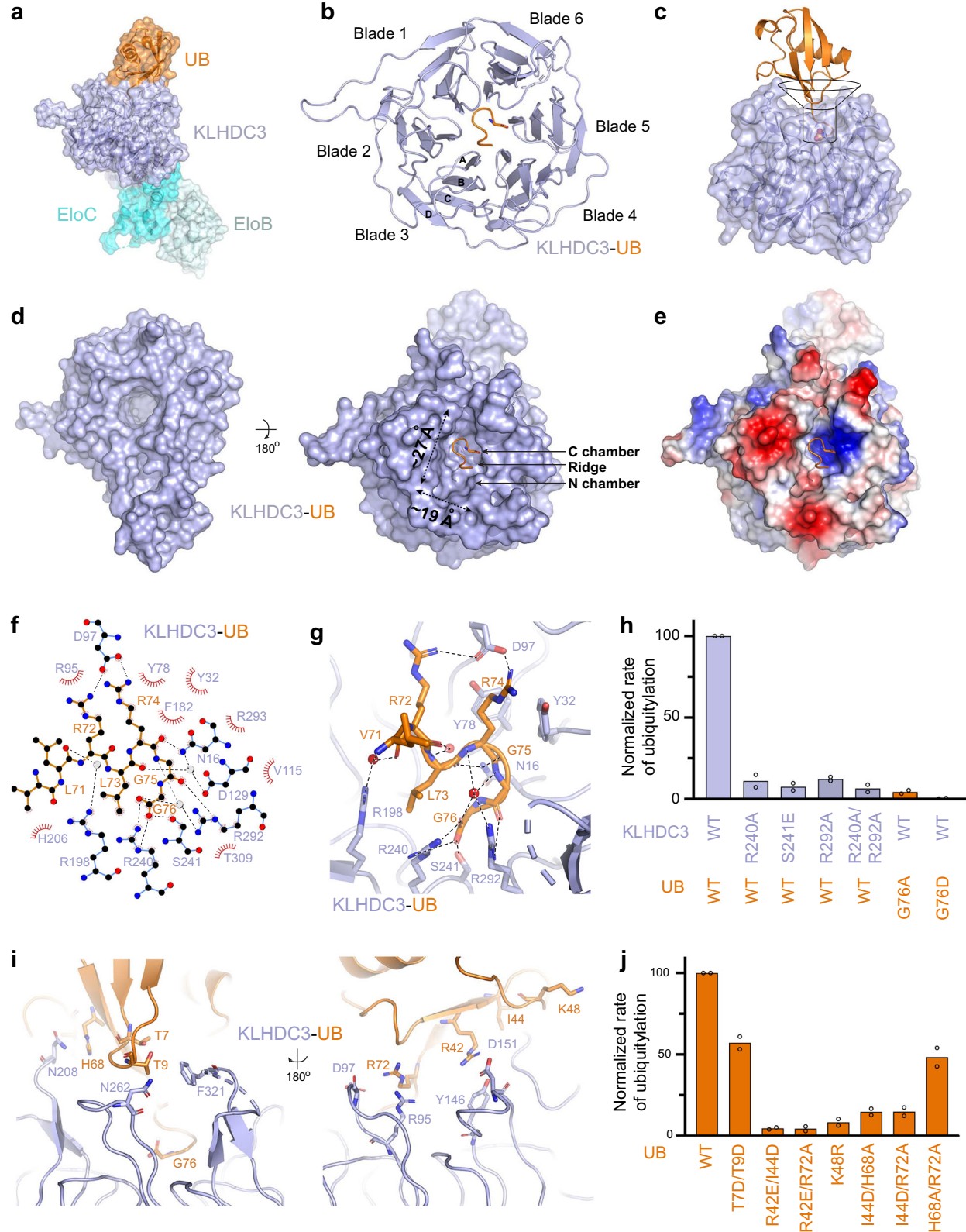

between KLHDC3 Asp97 and the side-chains of ubiquitin's Arg72 and Arg74.

Finally, loops connecting KLHDC3 form the top of the funnel-shaped substrate-binding site, and approach several residues from the globular domain of ubiquitin (Fig. 2i). Mutations to these ubiquitin residues reduced its capacity to serve as a KLHDC3 ubiquitylation substrate (Fig. 2j, Supplementary Fig. 2d–f).

## Ubiquitin substrate overcomes C-degron mimic autoinhibition of WT oligomeric KLHDC3

We previously showed that KLHDC2 and KLHDC3 are subject to C-degron-mimic mediated self-assembly[35]: the C-terminal Gly-Ser or His-Gly motifs from one KLHDC2 or KLHDC3 protomer, respectively, occupies the substrate-binding site of the adjacent protomer in a tetrameric assembly (Fig. 3a). Our extensive data for KLHDC2 revealed

**Fig. 2 | Structure of KLHDC3-EloB/C bound to a C-terminal Gly degron. a** Overall structure of UB bound to KLHDC3-EloB/C. UB (orange), KLHDC3 (light blue), EloC (cyan), and EloB (pale cyan) are shown in surface representation. **b** Cartoon representation of UB (orange) bound to KLHDC3 (light blue). UB's C-terminal Gly76 (orange) is shown in sticks. Blades 1–6 of the KLHDC3 propeller are labeled. The barrel beta-strands for one blade are labeled A-D. **c** Cartoon representation of the "funnel-like" arrangement of KLHDC3's substrate binding pocket. UB (orange) is shown in cartoon with its C-terminal Gly76 shown in spheres. KLHDC3 (light blue) is shown in cartoon with transparent surface. **d** Bottom (left) and top (right) views of KLHDC3 (light blue; surface). UB (orange,cartoon) is shown with its C-terminal Gly76 in sticks. The relative dimensions of the C-degron binding pocket and the C and N chambers of the KLHDC3 are labeled. **e** Electrostatic representation of KLHDC3 bound to UB (orange) with its C-terminal Gly76 in sticks. Hydrophobic surfaces are colored white, positively charged surfaces blue, and negatively charged surfaces red. **f** LigPlot of interactions between UB and KLHDC3, hydrogen bonds between KLHDC3 and UB or water (red, spheres) are show in black dashes. **g** Close up view of KLHDC3's (light blue) C-degron binding pocket. Residues from UB (orange) and KLHDC3 forming interactions are shown in sticks with black dashes between interacting partners. **h** Quantification of pulse-chase ubiquitylation assays monitoring UBE2R2 mediated ubiquitylation of UB or the indicated UB mutants by WT monomeric$c$ CRL2$^{KLHDC3}$ or the indicated KLHDC3 mutants. **i** Structural view of interactions between the globular domain of UB (orange) and KLHDC3 (light blue). Probable interacting residues are shown in sticks. **j** Quantification of pulse-chase ubiquitylation assays monitoring UBE2R2 mediated ubiquitylation of WT UB or UB globular domain mutants by monomeric CRL2$^{KLHDC3}$. Bar graphs are the average of $n = 2$ independent experiments. Source data are provided as a Source Data file.

that the self-assembly establishes a kinetic filter that enhances substrate selectivity beyond degron binding[35]. After a relatively short incubation period, bona fide substrates capture active monomeric KLHDC2-EloB/C during dynamic equilibrium between the tetrameric and monomeric species (Fig. 3a)[35]. However, proteins harboring C-termini matching a consensus degron, but that bind with a slow on-rate are rejected. Thus, although many proteins display C-degron consensus sequences and can bind an isolated KLHDCX substrate-binding domain, only a fraction have potential to serve as bona-fide substrates of WT KLHDC2 - and by extrapolation, KLHDC3-containing E3s[35]. Notably, we did not observe oligomerization of KLHDC1 or KLHDC10 complexes with EloB/C[35].

Since the biochemical assays employed for our initial substrate screening utilized monomeric versions of KLHDCX-EloB/C SRs, we assessed whether ubiquitin can serve as a substrate with the WT KLHDC3-EloB/C SR in combination with neddylated CUL2-RBX1. As observed previously for WT KLHDC2, both the monomeric mutant (G382K) and WT tetrameric KLHDC3 proteins are themselves subject to ubiquitylation in the absence of substrate (Fig. 3b)[35]. Remarkably, pre-incubation of ubiquitin (as a substrate) and wild-type KLHDC3-EloB/C for only two minutes was sufficient to promote its utilization as a ubiquitylation substrate at a level comparable to the monomeric KLHDC3 mutant (Fig. 3b). Thus, we surmise that ubiquitin is competent to capture active neddylated CRL2$^{KLHDC3}$ monomer generated from the equilibrium with the autoinhibited tetramer.

### KLHDC3 recognizes consensus C-degrons through a tripartite residue binding pocket

Since ubiquitin's C-terminal di-Gly motif is distinct from the KLHDC3 consensus degron sequence, we compared the affinity of monomeric KLHDC3-EloB/C for both ubiquitin and a peptide derived from the protein TCAP containing a consensus degron (Fig. 3c). Interestingly, the equilibrium dissociation constant ($K_d$) values were similar for ubiquitin and the TCAP-based peptide (Fig. 3c, Supplementary Fig. 3a, b, Supplementary Table 2; $K_d$ 321 nM and 224 nM, respectively). Mutation of ubiquitin's Gly75 to the consensus degron residues Arg or Gln resulted in ~3- and ~2-fold tighter binding, respectively (Fig. 3c, Supplementary Fig. 3d, e, Supplementary Table 2). Meanwhile, mutation of ubiquitin's Arg74 to Ala significantly diminished binding to monomeric KLHDC3-EloB/C, while combining Arg74Ala with a Gly75Arg mutation restored higher affinity binding to the SR complex (Fig. 3c, Supplementary Fig. 3d, f, Supplementary Table 2).

These results, including reduced affinity between Arg74Ala ubiquitin and monomeric KLHDC3-EloB/C, are consistent with prior results from a GPS-peptidome screen for KLHDC3-dependent degron sequences revealing a preference for Arg residues located at positions −2 through −6 of the degron[13]. In addition, our structure of KLHDC3 also shows interactions with the substrate ubiquitin's Arg72 and Arg74 (Fig. 2e). Thus, for KLHDC3, it appears that loss of the penultimate Arg in the degron can be compensated for by the presence of Arg residues located at positions −3 and −5. Interestingly, a chimera where ubiquitin's C-terminus was replaced with that from TCAP's showed more than an order-of-magnitude reduced affinity for monomeric KLHDC3-EloB/C (Fig. 3c, Supplementary Fig. 3g, Supplementary Table 2). This suggests that residues upstream of TCAP's C-terminal RG motif interact favorably with KLHDC3. The results from ubiquitylation assays containing neddylated CRL2$^{KLHDC3}$ and these substrates strongly correlated with substrate binding affinity to monomeric KLHDC3-EloB/C (Fig. 3d, Supplementary Fig. 3h, i).

To gain insights into the structural roles of penultimate residues within the degron sequences recognized by KLHDC3, the crystal structures of KLHDC3 bound to ubiquitin harboring Gly75Arg or Gly75Gln mutations were resolved to 2.6 and 1.9 Å resolution, respectively (Fig. 3e–g, Supplementary Fig. 3j, k, Supplementary Table 1). The structures superimposed with the wild-type ubiquitin-KLHDC3 complex with 0.23 and 0.09 RMSD over main chain atoms, respectively. Both mutant degrons engage the KLHDC3 binding pocket in a similar manner as for wild-type ubiquitin (Fig. 3e–g). The mutant ubiquitin's Arg or Gln side chains are accommodated by a KLHDC3 pocket consisting of Ser34, Thr309, and Asp325. In particular, Asp325 forms extensive interactions with side-chain atoms from the substrate ubiquitin's Arg or Gln (Fig. 3f, g, Supplementary Fig. 3j, k). To ascertain whether these interactions impact utilization as a substrate, wild-type ubiquitin, the Arg74Ala/Gly75Arg mutant, or ubiquitin-TCAP chimera proteins were tested as substrates in ubiquitylation assays with wild-type or mutant KLHDC3 proteins (Fig. 3h, Supplementary Fig. 3l–n). As expected, the Asp325Ala mutant KLHDC3 was relatively defective in comparison with the wild-type E3, but only towards substrates that contained an Arg residue in the penultimate position of the degron sequence (Fig. 3h, i, Supplementary Fig. 3l–n).

### Distinct structural features of C-terminal tail binding groove in KLHDC10

We next sought experimental insight into how KLHDC10 recognizes a protein's C-terminus. We did not obtain crystals of various KLHDC10 complexes with peptides or proteins. Thus, we employed our method of generating stable mimics of ubiquitylation transition states, which has enabled our attaining high-resolution structures of other CRLs by cryo-EM[25–28]. Although ARIH1 is not the preferred ubiquitin-carrying enzyme for this reaction (Fig. 1d, Supplementary Fig. 1c), we were able to obtain a cryo-EM structure at 3.33 Å resolution for the overall complex, and ~5.5 Å resolution over KLHDC10's propeller, representing ubiquitin transfer from the active site of neddylated CRL2$^{KLHDC10}$-activated ARIH1 to Lys48 on an acceptor ubiquitin recruited by KLHDC10 within the complex (Fig. 4a, Supplementary Fig. 4, Supplementary Table 3). The overall conformation of the ubiquitylation complex is similar to analogous complexes reported previously, and thus we focus here on substrate binding by KLHDC10. Visualization of details of the substrate binding site was achieved by focused refinement using a mask over the ubiquitin-KLHDC10-EloB/C region,

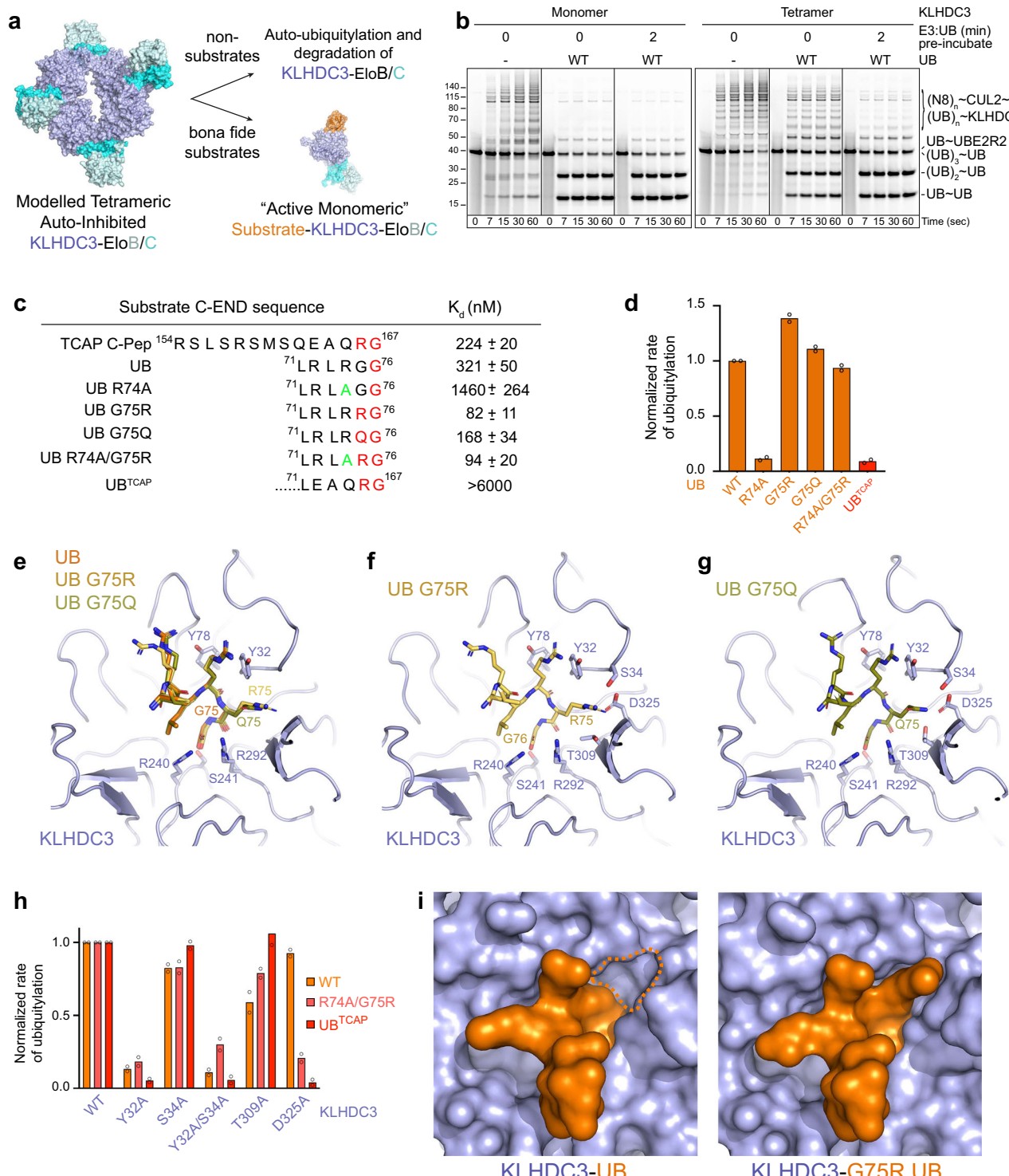

resulting in a map at 3.19 Å (Fig. 4a, Supplementary Figs. 4–6, Supplementary Table 3).

Like KLHDC2 and KLHDC3, KLHDC10 forms a six-bladed β-propeller (Fig. 4b)[34]. KLHDC10's propeller superimposes on that from KLHDC2 with 1.57 Å RMSD and from KLHDC3 with 1.15 Å RMSD over main chain atoms. Most strikingly, the C-terminal Gly adopts yet another unique orientation in KLHDC10: instead of facing blade 4 (KLHDC2) or 5 (KLHDC3) of the Kelch-type β-propeller, here it faces blade 6 (Fig. 4b).

The C-degron binding pocket within KLHDC10 is relatively smaller than in KLHDC2 or KLHDC3 (Fig. 4c; ~15 Å long by ~8 Å wide),

but maintains the organization of two chambers separated by a ridge (Fig. 4c)[34]. Differences between the arrangements of C-terminal binding amongst KLHDCX members can be viewed with the substrate-binding surface of the propeller facing upwards. This shows that unlike KLHDC2 and KLHDC3, the substrate ubiquitin's extreme C-terminal carboxyl group sits in a flat, side-to-side, conformation, as opposed to a more up and down arrangement observed with KLHDC2 and KLHDC3. Accordingly, mapping the electrostatic potential of KLHDC10's C-terminal carboxyl binding surface is consistent with fewer basic residues located in the binding chamber (Fig. 4d).

**Fig. 3 | Structural basis for penultimate Arg/Gln recognition in KLHDC3 degrons. a** Structural model depicting the presumed degron-mimic mediated auto-regulation of KLHDC3-EloB/C by the inactive tetrameric assembly and active monomer-substrate complexes derived from the KLHDC2-EloB/C tetrameric structure (8EBN.pdb). **b** Fluorescent scan of gel monitoring the ability of UB to serve as a substrate for the wild-type KLHDC3-EloB/C tetrameric assembly. Assays were performed in pulse-chase format with or without the indicated UB-E3 pre-incubation period prior to initiating ubiquitylation by addition of the UBE2R2 - UB thioester conjugate. **c** C-terminal sequences of the KLHDC3 degron peptide TCAP, WT UB, and the indicated C-terminal modifications of UB used in isothermal titration calorimetry and biochemical studies. Equilibrium binding affinities ($K_d$) are shown. Values are the average +/− 1 SD from $n = 2$ independent experiments. **d** Quantification of pulse-chase ubiquitylation assays monitoring ubiquitylation of UB and the indicated C-terminal UB modifications by monomeric CRL2$^{KLHDC3}$.

**e** Structural superposition of KLHDC3 (light blue, cartoon) bound to UB (orange, sticks), G75R UB (light orange, sticks), and G75Q UB (olive). KLHDC3's RSR motif, and its R72 and R74 UB interacting residues (Y78 and Y32) are shown in sticks. For clarity only KLHDC3 from the WT UB bound structure is shown. **f** Same as (**e**) but only for G75R UB. Potential KLHDC3 residues (S34, D325, and T309) for interaction with the penultimate residue in consensus degron sequences are shown in sticks. **g** Same as (**f**) but for G75Q UB. **h** Quantification of pulse-chase ubiquitylation assays monitoring UBE2R2 mediated ubiquitylation of UB or the indicated C-terminal UB modifications, by WT monomeric CRL2$^{KLHDC3}$ or the indicated KLHDC3 mutants. Bar graphs are the average of $n = 2$ independent experiments. **i** Close up view of the penultimate Arg binding pocket in KLHDC3 (light blue, surface). Wild-type UB (orange, surface) is shown with the penultimate pocket outlined in orange dashed lines (left). The right panel shows G75R UB (orange, surface) bound to KLHDC3 (light blue, surface). Source data are provided as a Source Data file.

Unexpectedly, KLHDC10 employs a variant motif - FSR (Phe366, Ser93, Arg391) - in blade 6 to engage the substrate's C-terminal carboxyl group (Fig. 4e, f). This differs from but is nevertheless related to the RSR motifs in blades 4 and 5 of KLHDC2 and KLHDC3, respectively. The substrate's carboxyl forms an anion-pi stacking interaction with KLHDC10's Phe366 in lieu of contacts with the first Arg from the RSR motif observed in KLHDC2 and KLHDC3. On the other hand, KLHDC10's Ser93 side-chain hydroxyl and Arg391 make interactions resembling those from the RSR motifs in KLHDC2 and KLHDC3.

The C-terminal di-Gly sequence is further anchored by KLHDC10's Ala155 from below and flanked by two aromatic residues, Tyr110 and Phe176, on one side opposite of Phe366. Furthermore, KLHDC10's Tyr208 interacts with backbone carbonyls from positions −3 and −4, while the backbone carbonyl from Phe176 interacts with ubiquitin's Arg72 side chain (Fig. 4e, f).

Notably, all KLHDC10 residues identified in our structure as interacting with ubiquitin's C-terminal sequence were previously shown by mutagenesis to impair Ala tail substrate binding in cells and by co-immunoprecipitation and pull-down studies with mutant KLHDC10 proteins[40]. KLHDC10's ability to recognize Ala tail substrates might be explained by a constellation of hydrophobic residues consisting of Leu127, Ala155, Try110, and Phe176 that from a small hydrophobic pocket adjacent to the Cα of Gly76, leaving just enough space to accommodate an Ala sidechain. Accordingly, biochemical ubiquitylation assays confirmed that KLHDC10 can accommodate degrons terminating in Ala, but a charged residue (Asp) is not tolerated (Fig. 4g, Supplementary Fig. 7a).

While the precise placement of ubiquitin's globular domain in the cryo-EM structure was challenging due to its flexibility, the arrangement visualized at low contour is reminiscent of the funnel-like organization of ubiquitin binding by KLHDC3 (Figs. 2c, 4h). This suggests that sidechain residues beyond the degron sequence might augment substrate recognition by KLHDC10. To test this, we subjected our panel of ubiquitin globular domain mutants to neddylated CRL2$^{KLHDC10}$ dependent ubiquitylation assays. Indeed, mutations to ubiquitin's globular domain hampered their ubiquitylation by KLHDC10 much like for KLHDC3 (Fig. 4i, Supplementary Fig. 7a–c).

### Recognition of consensus C-degron sequences by KLHDC10

KLHDC10 appears to prefer degrons terminating with the consensus sequence P/W-G[13,33]. However, inspection of the cryo-EM structure of KLHDC10 bound to ubiquitin showed insufficient room to accommodate a Pro or Trp residue (Fig. 5a). Specifically, KLHDC10's Phe176 side-chain stacks against the backbone of ubiquitin's penultimate C-terminal residue (Fig. 5a). However, structural modeling of an alternative rotamer for Phe176 revealed a potential small pocket adjacent to the substrate's penultimate residue that seemed capable of binding to hydrophobic side chains while restricting access to polar or charged side chains (Fig. 5b, c).

Surface Plasmon Resonance (SPR) binding studies were employed to estimate the affinity of KLHDC10 towards ubiquitin, ubiquitin tail chimeric proteins harboring the final five residues from consensus PG or WG degrons derived from COL24A1 or NFATC2IP, respectively, and a ubiquitin Gly75Trp mutant (Fig. 5d, Supplementary Fig. 7d–g, Supplementary Table 4). All the ubiquitin variants displayed similar binding affinities towards KLHDC10 ($K_d$ 250 nM–400 nM), with only marginal enhancement of binding for the ubiquitin-COL24A1 chimera and Gly75Trp ubiquitin. These results were further supported by ubiquitylation assays (Fig. 5e, f, Supplementary Fig. 7h, i).

### KLHDC3 and KLHDC10 E3s and UBE2R2 facilitate ubiquitin substrate ubiquitylation on a millisecond timescale

CRLs utilize one of numerous ubiquitin-carrying enzymes, each displaying distinct kinetics, to promote substrate ubiquitylation[23,26–28,30,48]. For example, UBE2D-family E2s and/or UBE2L3/ARIH1 are relatively efficient in attaching the first ubiquitin – i.e., "priming" - for substrates of CRL1 and CRL4-based E3s. On the other hand, UBE2R-family E2s were found to be relatively more efficient at substrate priming with some neddylated CUL2-based CRLs, and these E2s generally promote rapid extension of chains onto ubiquitin-primed substrates[27]. Interestingly, for our biochemical reactions with KLHDCX family E3s ubiquitin binding represents a unique case since the priming substrate itself is ubiquitin. To gain further insights, the ubiquitin-bound KLHDC3 complex was aligned onto our recently reported cryo-EM structures representing polyubiquitylation, which showed the basis for neddylated CRL2-dependent UBE2R2-mediated ubiquitylation of another ubiquitin (Fig. 6a)[27,28]. Ubiquitin presented by KLHDC3 could be accommodated by the activated complex (Fig. 6a). Indeed, mutation of the UBE2R2's acceptor ubiquitin binding site impaired KLHDC3 and KLHDC10-dependent modification of ubiquitin (Fig. 6b, c, Supplementary Fig. 8a, b), but the same mutation did not impact ubiquitylation of the substrate SELK (Fig. 6b, c, Supplementary Fig. 8a, b). Furthermore, mutation of acceptor UB's Arg54 to Asp dramatically reduced its rate of modification by KLHDC3 and KLHDC10 (Fig. 6d, Supplementary Fig. 8c). These results suggest ubiquitin could exhibit features of a poly-ubiquitin chain extension substrate even during priming by KLHDC3 and KLHDC10 E3s.

Thus, we queried if the rate of ubiquitin priming would be consistent with substrate priming by this E2 (0.1–1 s$^{-1}$), or poly-ubiquitin chain formation (20–100 s$^{-1}$). Pre-steady state kinetics performed on a quench flow instrument demonstrated that the rate of UBE2R2-catalyzed ubiquitin substrate priming with neddylated CRL2$^{KLHDC3}$ was remarkably fast (~50 s$^{-1}$) (Fig. 6e, g, Supplementary Fig. 8h). Although the rate of ubiquitin priming by neddylated CRL2$^{KLHDC10}$ was 2.65 s$^{-1}$ (Fig. 6f, g, Supplementary Fig. 8i) approximately 19-fold slower than the rate in combination with neddylated CRL2$^{KLHDC3}$, this is still amongst the fastest rates of substrate priming estimated for any CRL complex to date.

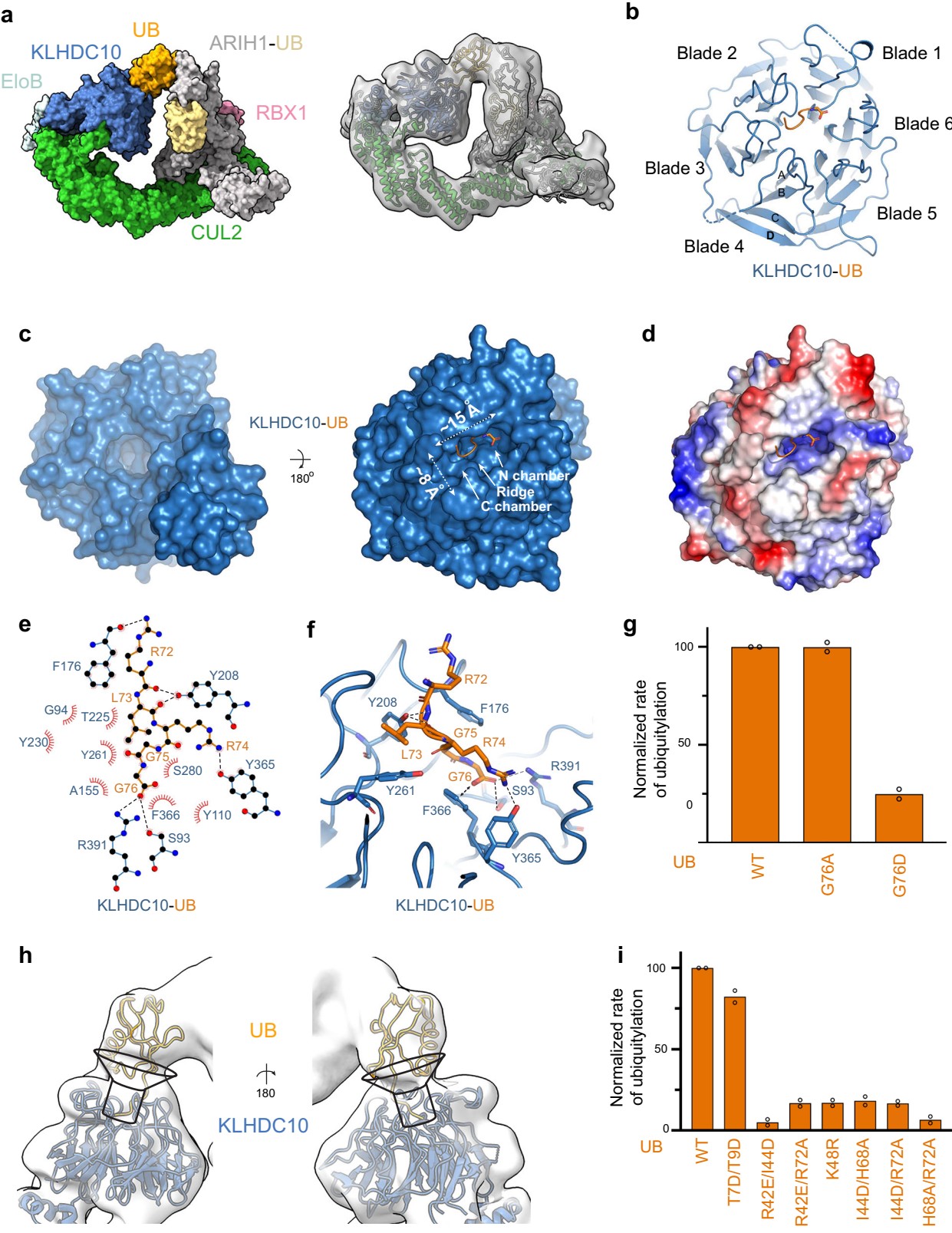

## Discussion

Here, by systematically comparing activities and experimentally-derived structures of a sub-family of CRL substrate receptors recognizing common degron features, we identified common and unique mechanisms determining their substrate specificities. Members of the KLHDCX family employ Kelch-type 6-bladed β-propellers to bind substrates, and a BC-box binding EloB/C that serve as bridges to a cullin (CUL2 or CUL5). Structures showed the top sides of the KLHDC2, 3, and 10 β-propellers all mediating substrate binding. Importantly, we discovered that these E3s share a defining C-terminal carboxylate-anchoring element (R/F-S-R) establishing them as C-degron E3s. Consistently, modeling the structure of KLHDC1 indicates conservation of the R/F-S-R motif for the entire family (Supplementary Fig. 9a).

**Fig. 4 | Structure of a trapped ARIH1-diUB-neddylated-CRL2^KLHDC10 complex reveals the basis for C-degron recognition by KLHDC10. a** Surface representation of the cryo-EM structure of a trapped transfer complex from neddylated-CRL2^KLHDC10 activated ARIH1 to Lys48 on a KLHDC10 bound acceptor ubiquitin (left). Structural coordinates from EloB/C-CUL2-RBX1 (cyan, pale cyan, green, and light pink respectively;5N4W.pdb), ARIH1-UB, (gray and wheat respectively;7B5M.pdb and 7B5N.pdb), UB-KLHDC10 (orange and marine; 1UBQ.pdb and an AlphaFold model of KLHDC10) were bulk fit into the EM density in ChimeraX (right). **b** Structure of UB (orange, cartoon) bound to KLHDC10 (marine, cartoon). UB's C-terminal Gly76 is shown in sticks. Blades 1–6 of the KLHDC10 propeller are labeled. The beta-strands for one blade are labeled A-D. **c** Bottom (left) and top (right) view of KLHDC10 (marine; surface). UB (orange) is shown in cartoon with its C-terminal Gly76 in sticks. The relative dimensions of the C-degron binding pocket

and the C- and N- chambers of KLHDC10 are labeled. **d** Electrostatic representation of KLHDC10 bound to UB (orange) with its C-terminal Gly76 in sticks. Hydrophobic surfaces are colored white, positively charged surfaces blue, and negatively charged surfaces red. **e** LigPlot of interactions between UB and KLHDC10. Hydrogen bonds between KLHDC10 and UB are show in black sticks. **f** Close up view of KLHDC10's (marine) C-degron binding pocket. Residues from UB (orange) and KLHDC10 forming interactions are shown in sticks with black lines. **g** Quantification of pulse-chase ubiquitylation assays monitoring UBE2R2 mediated ubiquitylation of UB or the indicated UB mutants by CRL2^KLHDC10. **h** Structural view of the funnel-like arrangement of KLHDC10's binding pocket. UB-KLHDC10 (orange and marine) are shown in cartoon within the cryoEM density at low contour. **i** Same as (**g**). Graphs are the average of $n = 2$ independent experiments. Source data are provided as a Source Data file.

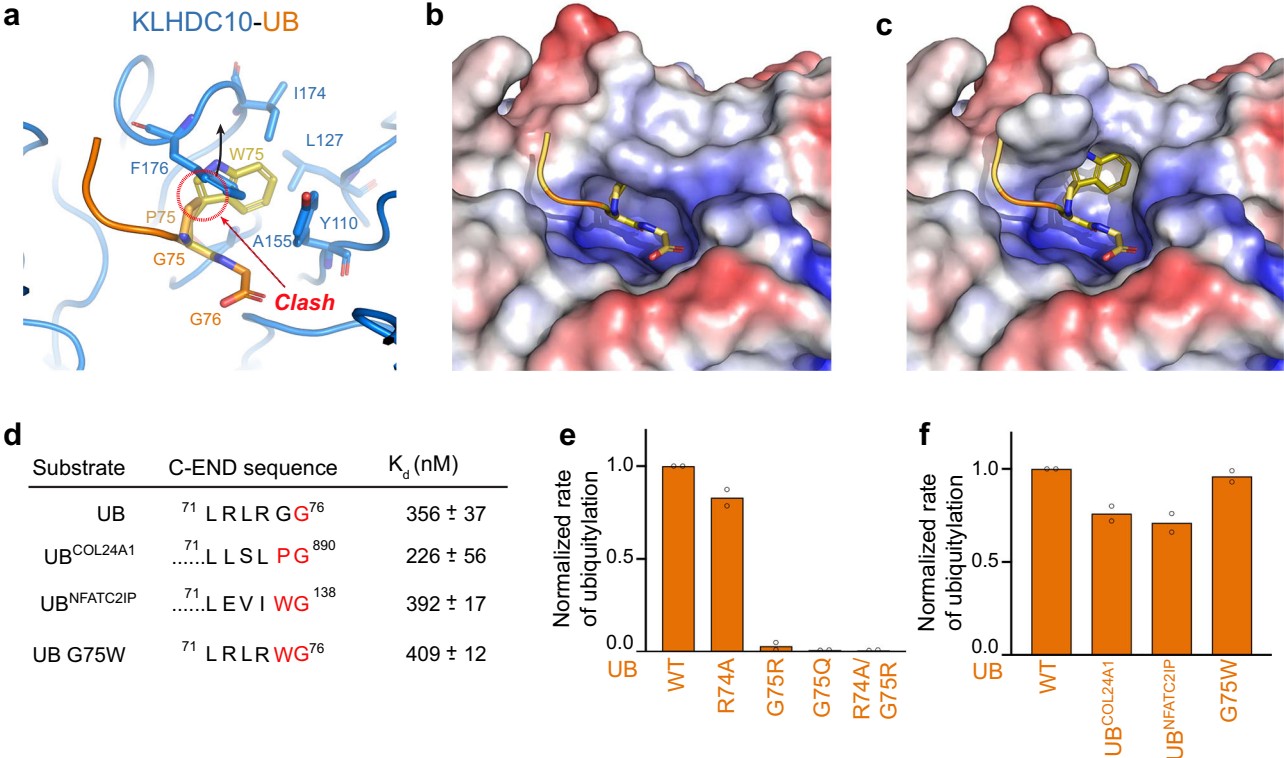

**Fig. 5 | Structural basis for penultimate residue tolerance amongst KLHDC10 degrons. a** Close up view of UB (orange) bound to the KLHDC10 (marine) C-degron binding pocket. Modeled conformations of consensus penultimate residues Pro (light orange, sticks) and Trp (yellow orange, sticks) are shown and their structural clash with KLHDC10 F176 is labeled. KLHDC10 residues (I174, L127, A155, and Y110) that form a cryptic hydrophobic pocket behind F176 are shown in sticks. **b** Surface representation of the KLHDC10 (marine) binding pocket, highlighting the structural clash of KLHDC10 with modeled penultimate residues Pro (light orange, sticks) and Trp (yellow orange, sticks). Hydrophobic surfaces are colored white, positively charged surfaces blue, and negatively charged surfaces red. **c** Same as

(**b**), but with a F176 flipped rotamer of KLHDC10 revealing a cryptic binding pocket that can accommodate penultimate Pro or Trp from KLHDC10 consensus degrons. **d** C-terminal sequences of the KLHDC10 degron variants used in surface plasmon resonance and biochemical studies. Equilibrium binding affinities obtained from SPR experiments ($K_d$) are shown. Values are the average +/– 1 SD from $n = 2$ independent experiments. **e** Quantification of pulse-chase ubiquitylation assays monitoring UBE2R2-mediated ubiquitylation of UB or the indicated C-terminal mutations by CRL2^KLHDC10. **f** Same as (**e**). Bar graphs are the average of $n = 2$ independent experiments. Source data are provided as a Source Data file.

Our data revealed three major principles of specificity for a KLHDCX E3's cohort of substrate C-degrons. First and unexpectedly, the spatially distinct locations of the carboxylate anchor elements in different propeller blades establishes different relative positioning of degron C-termini across the E3s (Fig. 7a–f). Importantly, each KLHDCX family member has only a single F/S-X-R motif, and different sequences in the other blades (for example, see Supplementary Fig. 9b–d). Second, the distinctly placed carboxylate anchor elements enable unique molecular environments specifying the C-degron profiles of an E3. For example, adjacent to the C-terminal Gly-binding site in KLHDC3, an aliphatic/acidic pocket determines preference for a Gly/C-degron's

penultimate Gln or Arg. The shape of the KLHDC3 pocket and acidic nature of adjacent surfaces explains this E3's general preference for Arg residues in substrate C-terminal regions. Meanwhile, analysis of the KLHDC10 structure together with biochemical data suggests how pliability of hydrophobic residues near the carboxylate anchor element contributes to this E3's capacity to recognize a range of C-degrons. While the KLHDC10 structure with ubiquitin's C-terminus shows how this E3 conforms to a Gly or Ala residue adjacent to substrate's C-terminal Gly (or Ala), altering the rotamer of an adjacent Phe (Phe176) would explain how the same E3 could display a distinct C-degron-binding groove that now prefers bulky and hydrophobic

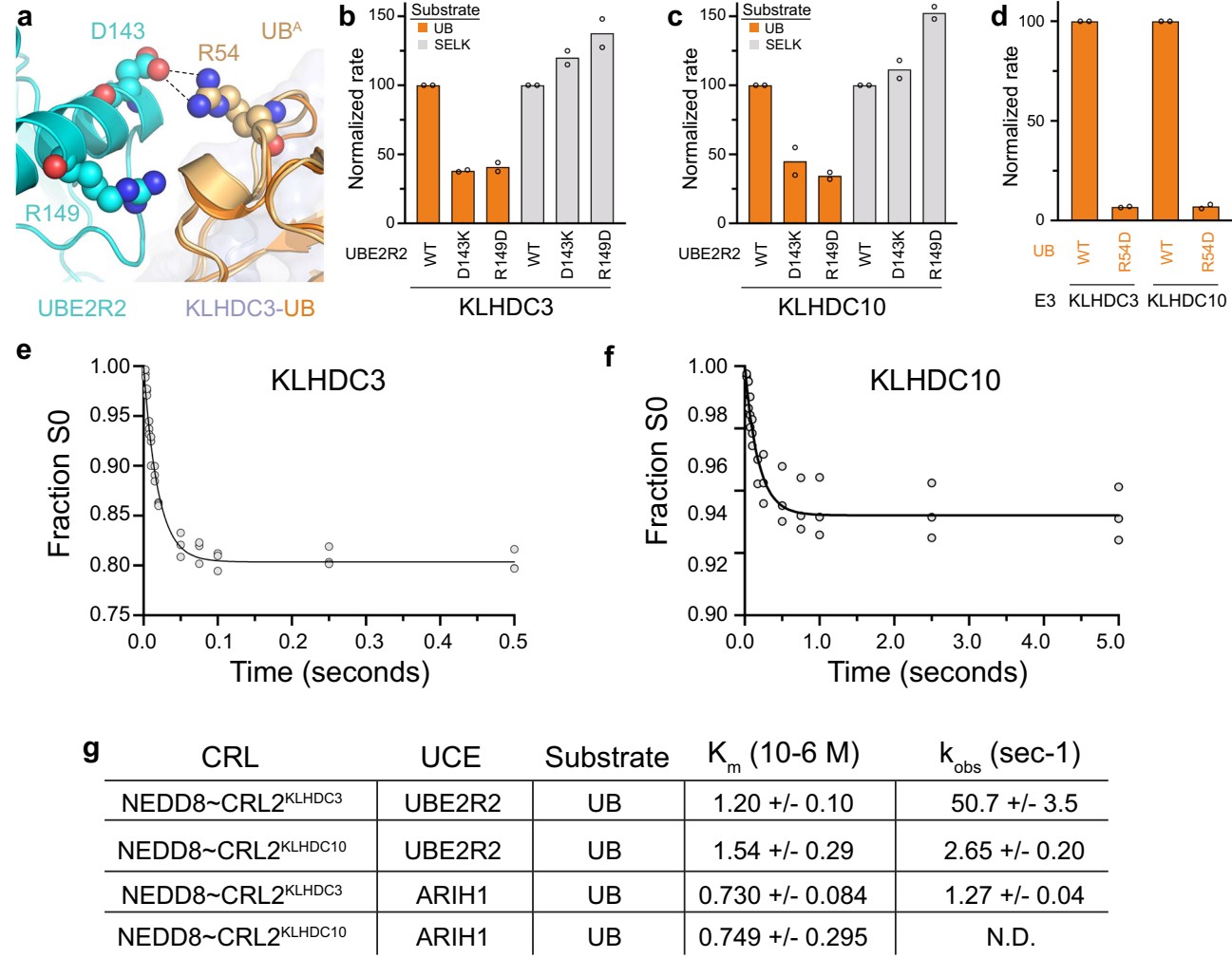

**Fig. 6 | Millisecond ubiquitylation of UB by CRL2^KLHDC3 and CRL2^KLHDC10.**
**a** Structural superposition of UB (orange) bound to KLHDC3 (light blue) to a UBE2R2 (cyan) acceptor UB (light orange) trapped polyubiquitylation structure (8PQL.pdb). Superpositions were generated by aligning UB from the KLHDC3 bound structure to the acceptor UB in the UBE2R2 trapped complex. UBE2R2 and acceptor UB residues making contacts are shown in spheres. **b** Quantification of pulse-chase ubiquitylation assays monitoring ubiquitylation of UB (orange) or SELK (gray) by WT or the indicated UBE2R2 mutants with monomeric neddylated-CRL2^KLHDC3. **c** Same as (**b**) but with neddylated-CRL2^KLHDC10. **d** Quantification of pulse-chase ubiquitylation assays monitoring ubiquitylation of WT UB or Arg54Asp UB by monomeric neddylated-CRL2^KLHDC3 or neddylated-CRL2^KLHDC10. Bar graphs are the average from $n = 2$ independent experiments. **e** Graphs from pre-steady state kinetic studies showing the fraction of UB substrate remaining as a function of time with monomeric neddylated-CRL2^KLHDC3. Data were fit to an analytical closed-form solutions model in Mathematica. Datapoints from triplicate technical replicates are shown in shaded circles. **f** Same as (**e**), but with neddylated-CRL2^KLHDC10. **g** Estimates of $K_m$ and $K_{obs}$ for UBE2R2 or UBE2L2/ARIH1 mediated ubiquitylation of UB by monomeric neddylated-CRL2^KLHDC3 and neddylated-CRL2^KLHDC10. Shown are the average +/− 1 SEM from $n = 3$ independent experiments. Source data are provided as a Source Data file.

penultimate side-chains. Our speculation that rotameric flipping of Phe176 is consistent with prior mutational analysis[40]. Finally, interactions distal from the C-degron binding site contribute to the recruitment and ubiquitylation of potential substrates. The importance of such interactions emerged from analysis of the KLHDC3-ubiquitin structure and effects of mutating residues in ubiquitin's globular domain - outside of the C-terminal degron region - that bind the propeller. Notably, unique loop sequences connecting strands within and between propeller blades contribute to unique surface topologies and features that could positively or negatively interact with potential substrates.

Furthermore, prior work showed how additional interactions could impact substrate selection. For instance, the C-termini of both KLHDC2 and KLHDC3 mimic their corresponding C-degrons, and mediate self-assembly into tetramers wherein the substrate-binding sites are blocked[35]. For KLHDC2, the rate of disassembly/assembly can serve as a kinetic filter for those substrates with fast enough on-rates to overcome reformation of the autoinhibited complex. However,

KLHDC10 appears to function without oligomerization, perhaps explaining why this substrate receptor employs a distinct mechanism to accommodate degron variability.

The principles that underlie KLHDCX family member degron recognition share commonalities with specificity determinants of other N- and C-degron specific ligases. For instance, consider the yeast multiprotein GID E3 complex, where two interchangeable substrate receptors (Gid4 and Gid10) share similar structures and recognize degrons containing an N-terminal Pro residue[53–59]. However, studies using phage display to explore interactions with non Pro/N-degron sequences have shown both the Gid4 and Gid10 degron binding pockets undergoing unique reshaping to conform to different N-termini and downstream sequences[60]. These features are conceptually analogous to the conformational change that would be required for KLHDC10 to recognize a diversity of partners. Gid4's substrate binding groove is also subject to remodeling by binding to yet another factor (Gid12)[61], which restricts binding to some substrates but not others. We speculate that factors may eventually be discovered

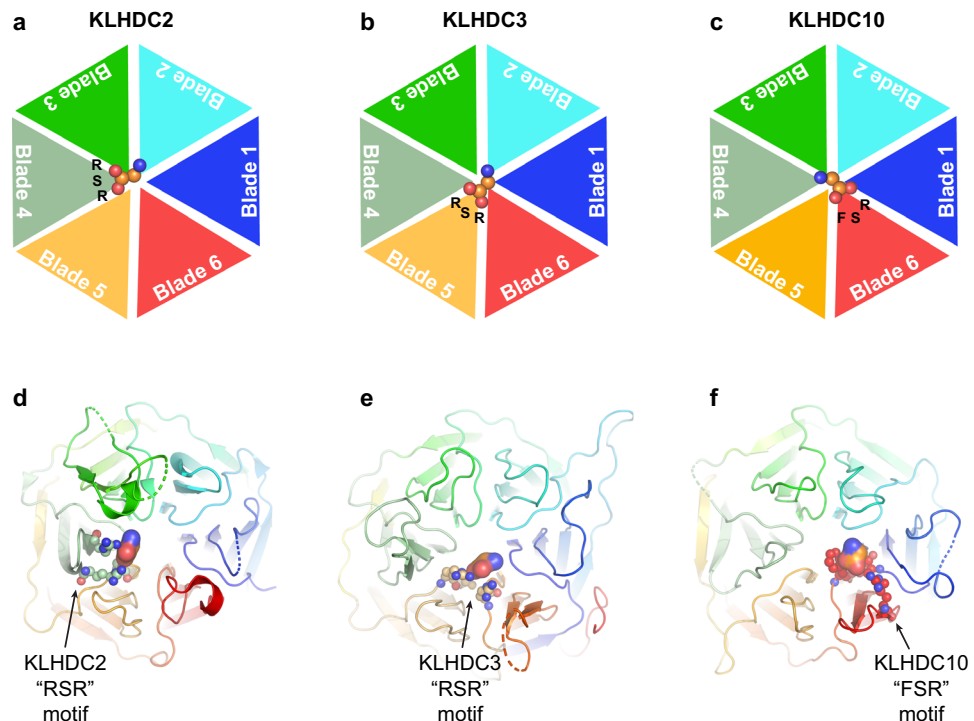

**Fig. 7 | KLHDCX family C-terminal anchors mediating C-degron placement and recognition. a** Cartoon representation of the arrangement of blades 1–6 from the β-propeller domain of KLHDC2. The relative positioning of the degron C-terminal carboxylate (orange, surface, colored by element) and its RSR recognition motif are shown. **b** Same as (**a**), but for KLHDC3s RSR motif. **c** Same as (**a**), but for KLHDC10s FSR motif. **d** Cartoon representation of the structure of KLHDC2 (rainbow, 8EBL.pdb) with the degron C-terminal carboxylate (orange, spheres, colored by element) and its RSR (pale green, surface, colored by element) recognition motif are shown. **e** Same as (**d**), but for KLHDC3 RSR motif (pale yellow, surface, colored by element). **f** Same as (**d**), but for KLHDC10 FSR motif RSR (red, surface, colored by element).

that affect the structures of KLHDCX family members that likewise reshape their degron binding pockets. Moreover, malleability and/or overlapping degron recognition are also conserved features of some other well-characterized families of N- and C-degron E3 ligases. For instance, ZER1 and ZYG11B both recognize distinct degrons containing a common N-terminal Gly residue through common folds[7,62-64]. Other C-degron ligases belonging to the FEM1A-C family members recognize degrons containing a C-terminal Arg residue[13,37], also through a common fold. In the case of FEM1B[65-68], extreme malleability is observed because it also contains an array of Cys and His residues that bind zinc ions with Cys/His-containing degrons[69,70].

Future studies will be required to determine if there are cellular circumstances when ubiquitin or UBLs are endogenous substrates of KLHDCX family E3s. In this regard, we note that unanchored poly-ubiquitin chains play roles in immune signaling and cellular stress responses[71]. Unanchored K48-linked poly-ubiquitin chains can modulate 26S proteasome function through competition with poly-ubiquitylated protein substrates[72,73]. In addition, the E3 ligase TRIM6 and E2 UBE2K forge unanchored K48-linked ubiquitin chains that stimulate antiviral defenses[74]. In terms of KLHDCX family members, we highlight two compelling observations. First, KLHDC3 and KLHDC10 show similar affinity for ubiquitin (~300 nM, Figs. 3c, 5d) as for endogenous KLHDCX E3 substrates substrates such as USP1. Second, ubiquitin linkage to ubiquitin is astonishingly fast, especially for KLHDC3 and UBE2R2, where the rate is comparable to biologically-relevant reactions.

Finally, the rules explaining C-degron recognition by KLHDCX family members provide insights into how this suite of E3s collectively contribute to the biological function of protein quality control. The embedding of specialized carboxylate anchor motifs within propellers mediates selection for C-termini at the end of flexible peptide-like sequences inadvertently produced by errors in translation or protein processing. Their distinct locations at the culmination of relatively rigid and/or malleable grooves, and broad range of distal surfaces displayed across a family establishes a range of interaction opportunities for such C-termini.

## Methods

### Cell lines

Tni cells were purchased from the Expression systems (94-002S) and maintained in suspension culture in ESF 921 media at 27 °C. Cells were not authenticated, but were routinely checked for mycoplasma contamination with LookOut mycoplasma PCR detection kit (Sigma).

### Constructs

Expression constructs generated for this study were prepared by standard molecular biology techniques and coding sequences entirely verified. Mutant versions used in this study were generated by QuickChange (Stratagene). Clones for bacterial expression of UBE2D2, UBE2M, UBE2F, UBE2R2, ARIH1, ARIH2, the NEDD8 E1 APPBP1-UBA3, NEDD8, UB, and SELK (residues 43-C representing the cytoplasmic domain of SELK) have been previously described[24-26,45,75]. Open reading frames for SUMO1, SUMO2, LC3A, LC3B, ISG15, FAT10, URM1, and URM1 were were obtained from Open Biosystems, PCR amplified and cloned into pGEX-2TK for bacterial expression. UB tail variants were constructed by PCR and cloned into a pGEX-4T1-TEV based vector.

Clones for insect cell expression of CUL2-RBX1, and CUL5-RBX2 have been previously described[24,35]. pBig1a based vectors for insect cell expression and purification of monomeric KLHDC2-EloB/C, monomeric KLHDC3-EloB/C and mutants thereof, full-length KLHDC3-EloB/

C, and KLHDC10[41-C]-EloB/C, have been described previously[35]. A clone for insect cell expression of KLHDC1-EloB/C consisted of first cloning full-length KLHDC1 (cDNA obtained from Open Biosystems) into pLib via Gibson assembly. Cassettes were then generated via PCR as described (Weissman) and Gibson assembled into pBig1a to generate a single vector for co-expression of all components. For SPR experiments a N-terminal 10X Histidine tag was placed on the amino-terminus of EloC by PCR and cloned into pLib via Gibson assembly. Cassettes were then generated via PCR as described (Weissman) and Gibson assembled into pBig1a to generate a single vector for co-expression of all components. Proper assembly of cassettes into pBig1a was confirmed by PmeI and SwaI restriction digestion.

## Protein expression and purification

UBE2D2, UBE2M, UBE2F, UBE2R2, UBE2L2, ARIH1, ARIH2, the NEDD8 E1 APPBP1-UBA3, UB tail variants, and SELK[43-C] were expressed in *E. coli* BL21 Gold (DE3) cells as GST fusion proteins. Fusion proteins were purified from cell lysates by glutathione affinity chromatography and liberated from GST by thrombin or TEV cleavage overnight at 4 °C. Cleavage reactions were further purified by ion exchange and size exclusion chromatography in 25 mM HEPES, 200 mM NaCl, 1 mM DTT pH = 7.5 (Buffer A). NEDD8, UB, SUMO1, SUMO2, LC3A, LC3B, ISG15, FAT10, URM1, and URM1 were expressed in *E. coli* BL21 Gold (DE3) cells as GST-thrombin fusion proteins. Fusion proteins were purified from cell lysates by glutathione affinity chromatography and liberated from GST by thrombin cleavage during extensive dialysis overnight in Buffer A at 4 °C. Cleavage reactions were passed back over a glutathione affinity resin to remove free GST and any remaining uncleaved GST-fusion protein. Protein collected in the flow fraction was concentrated with an Amicon Ultra filtration unit (3 K MWCO) and further purified by over a Superdex SD75 column in Buffer A.

Full-length CUL2-RBX1 was co-expressed in Tni insect cells as a His-Dac-TEV-CUL2 fusion with untagged RBX1 as previously described[35]. Fusion proteins were purified from cell lysates by Ni affinity chromatography and the His-Dac tag liberated by TEV cleavage overnight at 4 °C. Cleavage reactions were further purified by ion exchange and gel filtration over a Superdex SD200 column in Buffer A. Full-length CUL5-RBX2 was expressed in Hi5 insect cells as GST-TEV-RBX2 fusion along with untagged CUL5 as previously described[24]. Fusion proteins were purified from cell lysates by glutathione-sepharose affinity chromatography and the GST tag liberated from RBX2 by TEV cleavage overnight at 4 °C. Cleavage reactions were further purified by ion exchange and size exlusion chromatography in Buffer A.

Monomeric versions of KLHDC2-EloB/C, KLHDC3-EloB/C and mutants thereof, full length KLHDC1-EloB/C, full-length KLHDC3-EloB/C, KLHDC10[41-C]-EloB/C were expressed in Hi5 insect cells with a N-terminal 6XHis or 10XHis tag on EloC. Tagged proteins were purified from cell lysates by Ni affinity chromatography and further purified by ion exchange and gel filtration over a Superdex SD200 column in Buffer A.

Neddylation and purification of CUL2–RBX1, was prepared as described previously for CUL1-RBX1[45]. Briefly, the final concentrations of components in the neddylation reactions were as follows: 12 μM CUL2-RBX1, 1 μM UBE2M, 0.1 μM APPBP1-UBA3, and 20 μM NEDD8 in 25 mM HEPES, 200 mM NaCl, 10 mM MgCl2, 1 mM ATP, pH = 7.5. Reactions were initiated at room temperature by the addition of NEDD8 and incubated for ten minutes prior to quenching by the addition of DTT to 10 mM. Quenched reactions were spun at 13 K rpm for 10 min and immediately applied to a Superdex SD200 column to purify NEDD8-CUL2-RBX away from reaction components. Neddylation and purification of CUL5-RBX2 was performed as described above, with the exception that UBE2F was utilized as the NEDD8 E2 instead of UBE2M.

## Protein modifications

To introduce a cysteine for fluorescent labeling of UB and UBK0 (a lysine less variant of ubiquitin) we mutated the protein kinase A site in the pGEX2TK backbone converting the PKA site from RRASV to RRACV[45,76]. UB or UBK0 purified from this expression construct were labeled with AlexaFluor-647-Maleimide or Fluorescein-5-Maleimide respectively as previously described[45,76]. Briefly, DTT was added to UB or UBK0 at a final concentration of 10 mM and incubated on ice for 20 min to completely reduce cysteines for labeling. DTT was removed by buffer exchange over a NAP-5 column (GE Healthcare) in labeling buffer (25 mM HEPES, 200 mM NaCl). Labeling reactions consisted of UB or UBK0 at 150 μM final concentration and were initiated by the addition of 600 μM AlexaFluor-647-Maleimide or Fluorescein-5-Maleimide (4X excess over labeling target and <5% final DMSO concentration). Reactions were incubated at room temperature for 2 h and quenched by the addition of DTT to 10 mM. Quenched reactions were desalted over a PD-10 column in labeling buffer containing 1 mM DTT to remove unreacted probe. Desalted protein was concentrated in an Amicon Ultra filtration unit (3 K MWCO) and further purified over a Superdex SD75 column.

## Crystallography

For the structure of monomeric KLHDC3-EloB/C bound to ubiquitin, the complex was first purified by size exclusion chromatography in Buffer A by loading a pre-equilibrated mixture of KLHDC3-EloB/C (40 μM final) with UB (160 μM final). Protein fractions were pooled and concentrated to ~8 mg/ml with an Amicon Ultra filtration unit (10 K MWCO). Crystals grew as clusters at 4 °C in 8–10% PEG5KMME, 5% Tacsimate pH 7.0, 0.1 M HEPES pH = 7.0. Diffraction quality crystals were prepared by streak seeding in 5–7% PEG5KMME, 5% Tacsimate pH 7.0, 0.1 M HEPES pH = 7.0. Crystals were harvested in mother liquor supplemented with 25% Glycerol or 25% MPD prior to flash-freezing in liquid nitrogen. Reflection data for the WT complex was collected at SERCAT 22-BM at the Advanced Photon Source, while data for G75Q and G75R mutant UB complexes were collected at the ALS-5.0.1 at the Advanced Light Source.The crystals belong to space group C222₁ with one UB-KLHDC3-EloB/C complex in the asymmetric unit. Phases were obtained by molecular replacement using PHASER[77] searching for one copy of an AlphaFold model of KLHDC3, and one copy of EloB/C from 1VCB.pdb. Manual building was performed in COOT[78] and refinement was performed using Phenix[79]. Additional details of the refinement are provided in Supplementary Table 1.

## Biochemical assays

The use of pulse-chase assays allowed comparing the paths of UB transfer starting from either UBE2D2, UBE2R2, or UBE2L2. First, the indicated E2 was pulse-labeled by incubating a mixture of UBA1 (0.3 μM), E2 (10 μM), and fluorescently labeled UB (15 μM) in 25 mM HEPES, 100 mM NaCl, 100 mM MgCl2, ATP (2 mM), pH 7.5 at room temperature for 13 min. Pulse-loading reactions were quenched by the addition of EDTA to 50 mM and incubated on ice for 5 min. Chase reactions consisted of mixing the E2 ~ UB thioester conjugate (0.3 μM final concentration) with the indicated pre-equilibrated NEDD8 ~ CUL2-RBX1-SR complexes (0.3 μM final concentration) UBL (0.8 μM final concentration) in 25 mM HEPES, 100 mM NaCl, 50 mM EDTA, 0.5 mg/ml BSA, pH 7.5 at room temperature. Where indicated test substrates were pre-incubated (0.6 μM final concentration) with NEDD8 ~ CUL2-RBX1-SR complexes for the indicated times prior to initiating reactions by the addition of UBE2R2 ~ UB. Reactions for Fig. 1b−e, Supplementary Fig. 1a−d, and Fig. 3b were performed with wild-type UB loaded UCEs. Samples in Fig. 1b−e and Supplementary Fig. 1a−d were quenched at the indicated times with 2X SDS-PAGE sample buffer supplemented with 50 mM DTT to reduce the UBE ~ UB thiolester conjugate to aid in product quantifications. Quenched

samples were separated on 4–12% Bis-Tris gradient gels and scanned for fluorescence on a Typhoon imager. For subsequent pulse-chase assays it was necessary to slow the reactions down in order to observe product formation over time for activity comparisons. This was accomplished by performing pulse-chase assays as descrived above but with KLHDC3-EloB/C reactions were performed in MES buffer pH = 6.5 at 0 °C. Similarly, reactions containing KLHDC10-EloB/C were performed in HEPES buffer pH = 7.5 at 0 °C. We also utilized UBE2R2 ˜ UBK0 in these assays to prevent polyubiquitylation chain formation.

## Wild-type ubiquitin substrate radiolabeling

All radiolabeling reactions contained 16 μM [γ-$^{32}$P]ATP, 100 μM wild-type ubiquitin substrate, and protein kinase A in 1X NEBuffer™ for Protein Kinases (PK). The reaction was incubated for 1 h at 32 °C before adding 100 μM cold ATP to ensure all wild-type ubiquitin substrate is modified with a phosphate group. After this, the reaction was incubated for one more hour at 32 °C.

## Estimation of the $K_m$ of wild-type UBE2R2 for CRL2$^{KLHDC3}$ and CRL2$^{KLHDC10}$ under multi-turnover conditions

In these multi-turnover ubiquitylation reactions, the concentration of radiolabeled wild-type ubiquitin substrate was ten-fold greater than the concentration of the CRL2 complexes. Purified proteins were sequentially diluted in two separate Eppendorf tubes containing reaction buffer (30 mM Tris, pH 7.5, 100 mM NaCl, 5 mM MgCl$_2$, 2 mM DTT, and 2 mM ATP) at room temperature (20 °C). The "CRL mix" was assembled in one tube by addition of neddylated CUL2-RBX1 (0.5 μM), the substrate receptor KLHDC3 or KLHDC10 (0.5 μM), and radiolabeled wild-type ubiquitin substrate (5 μM). While the components of the CRL mix were incubating, the "E1 mix" was assembled in the second tube by addition of E1 ubiquitin-activating enzyme (0.625 μM) and unlabeled K48R ubiquitin donor (75 μM). After a brief one-minute incubation period, the E1 mix was evenly aliquoted into nine Eppendorf tubes. Next, dilutions of wild-type UBE2R2 from a two-fold serial dilution series were added to the nine aliquots to initiate UBE2R2 charging with K48R donor ubiquitin (UBE2R2 concentrations after addition ranged from 35.4 μM to 138 nM). After a 15-min incubation period, reactions were initiated by addition of an equal volume of CRL mix to each tube containing charged UBE2R2 protein. Following a 10-s reaction period, each reaction was quenched by addition of an equal volume of reducing 2X SDS-PAGE loading buffer (100 mM Tris, pH 6.8, 30 mM EDTA, pH 8.0, 20% glycerol, 4% SDS, and 4% β-mercaptoethanol). Substrate and product were resolved on 18% Tris-Glycine SDS-PAGE gels. Autoradiography was performed using an Amersham Typhoon 5 imager and quantification of the resolved protein bands was accomplished using ImageQuant software. The fraction of substrate ubiquitylated was calculated as the signal of product (defined as a substrate that has been modified by at least one donor ubiquitin) divided by the total signal (product and substrate). The "fraction ubiquitylated" values were plotted as a function of the UBE2R2 concentrations on a graph by fitting the data to the Michaelis-Menten model using nonlinear regression (GraphPad Prism v10).

## Pre-steady state single-encounter ubiquitylation reactions of wild-type UBE2R2 in the presence of CRL2$^{KLHDC3}$ and CRL2$^{KLHDC10}$

In these single-encounter reactions, unlabeled SelK peptide was always present at a final concentration of 25 μM which was 100-fold greater than radiolabeled wild-type ubiquitin substrate. Purified proteins were sequentially diluted in two separate Eppendorf tubes containing reaction buffer at room temperature. The "CRL mix" was assembled in one tube as described in the previous section except radiolabeled wild-type ubiquitin substrate was 0.5 μM. While the components of the CRL mix were incubating, the "E1 mix" was assembled in the second tube by addition of E1 ubiquitin-activating enzyme (0.5 μM) and unlabeled

K48R ubiquitin donor (30 μM). After a brief one-minute incubation period, wild-type UBE2R2 (24 μM) was added to the E1 mix to initiate UBE2R2 charging. The contents of the CRL mix were then loaded into one sample loop of a KinTek RQF-3 Quench Flow. Following a 15-minute incubation period to ensure complete charging of UBE2R2, unlabeled SelK peptide (50 μM) was added to the E1 mix and then the contents of the E1 mix were loaded into a separate sample loop of the Quench Flow. Reactions were initiated by bringing the two protein mixes together in drive buffer (30 mM Tris, pH 7.5 and 100 mM NaCl) and then quenched at various time points in reducing 2X SDS-PAGE loading buffer. Substrate and product were resolved on 18% Tris-Glycine SDS-PAGE gels. Autoradiography was performed using an Amersham Typhoon 5 imager and quantification of the protein bands was accomplished using ImageQuant software. The fraction of substrate ubiquitylated was calculated as the signal of product (defined as a substrate that has been modified by at least one donor ubiquitin) divided by the total signal (product and substrate). The rates of ubiquitin transfer by UBE2R2 were estimated by fitting the data to an analytical closed-form solutions model in Mathematica (v13.1).

## Estimation of the $K_m$ of wild-type ARIH1 for CRL2$^{KLHDC3}$ and CRL2$^{KLHDC10}$ under single-encounter conditions

In these single-encounter reactions, unlabeled SelK peptide was always present at a final concentration of 25 μM which was 100-fold greater than radiolabeled wild-type ubiquitin substrate. Purified proteins were sequentially diluted in two separate Eppendorf tubes containing reaction buffer at room temperature. The "CRL mix" was assembled in one tube as described in the previous section. While the components of the CRL mix were incubating, the "E1 mix" was assembled in the second tube by addition of E1 ubiquitin-activating enzyme (1.67 μM) and unlabeled K0 ubiquitin donor (67 μM). After a brief one-minute incubation period, wild-type UBE2L3 (50 μM) was added to the E1 mix. After a two-minute incubation period to ensure complete charging of UBE2L3, the E1 mix was evenly aliquoted into nine Eppendorf tubes. Next, dilutions of wild-type ARIH1 from a two-fold serial dilution series were added to the nine aliquots (ARIH1 concentrations after addition ranged from 15 μM to 59 nM). After a one-minute incubation period, unlabeled SelK peptide (50 μM) was added to each aliquot. Reactions were then initiated by addition of an equal volume of CRL mix to each tube containing ARIH1 protein. After a 10-s reaction period, each reaction was quenched by addition of an equal volume of reducing 2X SDS-PAGE loading buffer. Substrate and product were resolved on 18% Tris-Glycine SDS-PAGE gels. Autoradiography was performed using an Amersham Typhoon 5 imager and quantification of the protein bands was accomplished using ImageQuant software. The fraction of substrate ubiquitylated was calculated as described in the previous sections. The fraction ubiquitylated values were plotted as a function of the ARIH1 concentrations on a graph by fitting the data to the Michaelis-Menten model using nonlinear regression (GraphPad Prism v10 software).

## Pre-steady state single-encounter ubiquitylation reactions of wild-type ARIH1 in the presence of CRL2$^{KLHDC3}$ and CRL2$^{KLHDC10}$

In these single-encounter reactions, unlabeled SelK peptide was always present at a final concentration of 25 μM which was 100-fold greater than radiolabeled wild-type ubiquitin substrate. Purified proteins were sequentially diluted in two separate Eppendorf tubes containing reaction buffer at room temperature. The "CRL mix" was assembled in one tube as described in the previous section. While the components of the CRL mix were incubating, the "E1 mix" was assembled in the second tube by addition of E1 ubiquitin-activating enzyme (0.5 μM) and unlabeled K0 ubiquitin donor (12.5 μM). After a brief one-minute incubation period, wild-type UBE2L3 (10 μM) was added to the E1 mix. After a two-minute incubation period to ensure complete charging of UBE2L3, wild-type ARIH1 (5 μM) was added to the E1 mix followed by

unlabeled SelK peptide (50 µM). While the E1 mix was incubating, the contents of the CRL mix were loaded into one sample loop of a KinTek RQF−3 Quench Flow. Then, the contents of the E1 mix were loaded into a separate sample loop of the Quench Flow. Reactions were initiated by bringing the two protein mixes together in drive buffer and then quenched at various time points in reducing 2X SDS-PAGE loading buffer. Substrate and product were resolved on 18% Tris-Glycine SDS-PAGE gels. Autoradiography was performed using an Amersham Typhoon 5 imager and quantification of the protein bands was accomplished using ImageQuant software. The fraction of substrate ubiquitylated was calculated as described in the previous section. The rates of ubiquitin transfer by ARIH1 were estimated by fitting the data to an analytical closed-form solutions model in Mathematica (v13.1). While pre-steady state single-encounter reactions were attempted with ARIH1 and neddylated CRL2(KLHDC10), product formation was too inefficient to enable estimation of the rate of ubiquitin transfer from ARIH1 to labeled ubiquitin substrate.

## Surface plasmon resonance
SPR was utilized to determine kinetic parameters and binding affinity of KLHDC10 substrates using a Cytiva Biacore T200. Experiments were carried out in 25 mM Tris, 200 mM NaCl, 0.1% Tween20, at pH 7.4 as a running buffer with a flow rate of 80 µL/min at 20 °C.10xHis-KLHDC10 was diluted to 25 µg/mL in running buffer and immobilized onto a Ni-NTA-coated biosensor series S Cytiva chip leading to an average capture of 300 Rus per cycle. The chip surface was conditioned according to the manufacturer's instructions. The surface was then activated using an injection of 0.5 M EDTA (125 µL at 25 µL/min) followed by an injection of 0.5 mM NiCl$_2$ (125 µL at 25 µL/min). KLHDC10 was then immobilized onto channel 2 or channel 4 with appropriate reference channels as described next. Reference channels (fc1 and fc3) were treated identically to channels 2 and 4, except for protein loading, to be used as a reference channel. Protein analyte substrates were diluted to their designated concentrations in the running buffer and injected with a flow rate of 80 µL/min at 20 °C. Protein substrates were injected with a top concentration of both 2 µM followed by a 1:2 serial dilutions to 0.125 µM series. The concentration series of each analyte was injected in a multi-cycle kinetics mode, at 80 µL/min flow rate, with 120 s association and 800 s final dissociation phase followed by a regeneration of 120 s with 350 mM EDTA to reach baseline after protein–protein interaction. All samples were run in duplicate. Double-referenced sensorgrams were analyzed using 1:1 binding model, which fit data (Biacore Evaluation Software v.3.1).

## Isothermal titration calorimetry
All experiments were performed on a MicroCal AutoITC200. Proteins were first buffer matched by desalting over a NAP5 column in ITC buffer (25 mM HEPES, 150 mM NaCl, 1 mM BME, pH = 7.5). Protein concentrations were then determined by nanodrop. KLHDC3-EloB/C was diluted to 12 µM in ITC buffer and placed in the sample cell. Wild type Ubiquitin or mutant variants were diluted to 250 µM and placed in the sample syringe. Titrations were performed at 22 °C with one injection of 0.4 µl, followed by 13 injections of 3 µl.

## Cryo-EM grid preparation and data acquisition
Activity-based probes (ABPs) were used to mimic the native intermediate of donor ubiquitin transfer to CRL substrate-linked acceptor ubiquitin by ARIH1. The ABP used His-tagged-ubiquitin(1−75)−MESNa and its conjugation to the compound (E)-3-[2-(bromomethyl)−1,3-dioxolan-2-yl]prop-2-en-1-amine (BmDPA) as previously described[25]. Reactive His-ubiquitin(1−75)−BmDPA (which mimics the donor ubiquitin in the final trapped complex; 0.5 mg ml⁻¹ final) was incubated with 100 µM K48C ubiquitin for 1 h at 30 °C. The ABP was purified by size-exclusion chromatography in a column that had been equilibrated with a buffer containing 25 mM HEPES pH 7.5 and 150 mM NaCl.

To form the trapped complex (containing neddylated CRL2^KLHDC10, open-mutant A ARIH1 was incubated with 1 mM TCEP for 20 min on ice and desalted (Zeba spin columns) into a buffer that contained 25 mM HEPES pH 7.5 and 150 mM NaCl. Next, 10 µM desalted ARIH1 was immediately added to other complex components including 5 µM neddylated CRL2^KLHDC10 and a sixfold molar excess of ABP. The reactions were incubated for 30 min at 30 °C. The trapped complex was purified by size-exclusion chromatography on a column that had been equilibrated in a buffer containing 25 mM HEPES pH 7.5, 75 mM NaCl and 1 mM TCEP.

For cryo-EM grid preparation, 3 µl of the purified ARIH1-diUB-neddylated-CRL2^KLHDC10 complex at 4 µM protein concentration was applied onto UltraAufoil 0.6/1 300-meshgrids (Quantifoil) freshly plasma cleaned with Ar/O$_2$ gas mixture for 30 s using Solarus plasma cleaner (Gatan). The sample was immediately blotted for 4 s followed by plunge freezing into liquid ethane cooled to liquid N$_2$ temperature with a Vitrobot Mark IV (ThermoFisher Scientific, TFS) operating at 6 °C and 100% humidity.

Cryo-EM imaging was performed using a 300 kV Titan Krios G3i (TFS) transmission electron microscope equipped with K3 detector (Gatan) and a post-column BioQuantum Imaging filter (Gatan) with a slit width set to 20 eV. K3 gain reference was collected prior to data collection. A total of 13,395 movies (binned x2) were recorded in super-resolution counted mode at 130,000x magnification (physical pixel size of 0.6485 Å) over a defocus range of −0.6 to −2.25 µm using EPU automated image acquisition software (TFS). Each exposure was dose fractionated into 40 frames collected over 1.2 s with a dose rate of 13.95 e⁻/px/s and a total dose of ~40 e⁻/Å².

## Cryo-EM image processing
All steps of cryo-EM image processing of ARIH1-diUB-neddylated-CRL2^KLHDC10 dataset were performed in the framework of cryoSPARC v4.2.1[80] (Supplementary Fig. 4). The full dataset of 13,395 movies were subjected to motion correction using Patch Motion Correction followed by estimation of contrast transfer function (CTF) using Patch CTF Estimation jobs as implemented in cryoSPARC (Supplementary Fig. 4). At this stage, 3365 micrographs were discarded based on estimated average defocus value higher than −2.6 µm and CTF fit resolution worse than 6 Å. Out of the remaining 10,030 micrographs, an initial set of 247 particles were manually picked from 13 micrographs. These manually picked particles were used as an input for Blob Tuner job for picking particles for all the accepted micrographs. The picked particles (1,554,802) were extracted using a box size of 720 pixels (~467 Å) and Fourier cropped to 360 pixels. The extracted particles (1,055,840) were subjected to 2D classification (k = 200, maximum resolution: 8 Å) to remove junk or inconsistent particles. The remaining particles (337,866) were subjected to Ab-initio Reconstruction with three classes (highlighted by the numbers in Supplementary Fig. 4). One of these classes (160,418 particles) corresponding to the fully assembled complex was homogeneously refined to reveal secondary structure features. To be noted that the hand of the homogeneously refined map was Z-flipped to generate the map with the correct hand. However, for clarity, we show the correct hand maps for the selected ab-initio class (class 1), as well as for the homogeneously refined map, in Supplementary Fig. 4. To further reduce any heterogeneity, all 337,886 particles retained after the 2D classification i.e., particles from the homogeneously refined map (160,418) combined with those from ab-initio classes 2 (93,477) and 3 (83,991), were used as input for heterogeneous refinement (Refinement box: 360 pixels). Two 3D volumes were employed as initial references: the Z-flipped map from homogeneous refinement and the class 3 from ab-initio reconstruction. (Supplementary Fig. 4). The class corresponding to the bonafide ARIH1-diUB-neddylated-CRL2^KLHDC10 complex (class 1) was further subjected to Non-uniform Refinement followed by Local Refinement

(using pose/shift gaussian prior during alignment) to obtain the final high-resolution consensus map at an overall resolution of 3.3 Å.

In an attempt to improve the local resolution of the KLHDC10-EloB/C region of the consensus map (Supplementary Fig. 5d, e), further processing of the dataset was pursued using particle subtraction approach. The region of the consensus map excluding the KLHDC10-EloB/C components (Supplementary Fig. 4, shown in transparent white) was used for particle subtraction, followed by Local Refinement (using pose/shift gaussian prior during alignment) of the subtracted particles using a mask (Supplemenetary Fig. 4, shown in transparent pink) encompassing the KLHDC10-EloB/C region. These attempts yielded a focused map corresponding to KLHDC10-EloB/C region at an overall resolution of 3.2 Å (Supplementary Fig. 6).

The Fourier shell correlation (FSC) curves were calculated in cryoSPARC, and reported resolutions are based on the gold-standard 0.143 criterion. Local resolution calculations were performed in cryoSPARC. Supplementary Figs. 5, 6 shows representative analysis corresponding to consensus and focused maps, respectively, reported in this study. Supplementary Table 3 summarizes essential cryo-EM data collection and processing statistics.

### Model building and structure refinement

An initial model corresponding to KLHDC10-EloB/C focused map was generated in ChimeraX using rigid-body fitting of the coordinates available for previously reported crystal structure of EloB/C complex (PDB: 1VCB) and the alpha-fold model of KLHDC10 (Entry: Q6PID8) (Supplementary Fig. 6). To be noted that the N-terminal region of KLHDC10 is largely not observed in our cryo-EM map, however, we observed additional density that unambiguously accommodated the segment constituted by N41-G50 of the KLHDC10 alpha-fold model. In addition to KLHDC10-EloB/C, the focused map also revealed an additional segment of density in the degron binding groove of KLHDC10. We attribute this density to ubiquitin c-degron (73-LRGG-76), in agreement with our biochemical analysis as well as cryo-EM studies that shows density for the UB bridging the KLHDC10 and ARIH1 regions in the consensus map. Main-chain and/or side-chain atoms were removed for residues with missing or poorly resolved density. The initial model was iteratively optimized by manual building using COOT[78] and real space refinement using PHENIX[79]. Supplementary Table 3 summarizes essential structure refinement and validation statistics. Figures were prepared using UCSF ChimeraX[81].

### Reporting summary

Further information on research design is available in the Nature Portfolio Reporting Summary linked to this article.

## Data availability

The X-ray crystallography data have been deposited in the RCSB with accession codes 9D1I (KLHDC3-EloB/C bound to ubiquitin), 9D1Y (KLHDC3-EloB/C bound to G75R ubiquitin), 9D1Z (KLHDC3-EloB/C bound to G75Q ubiquitin), 9D8P (focused map: UB-KLHDC10-EloB/C) and in the Electron Microscopy Data Bank with codes EMDB: EMD-46644 (consensus map:ARIH1-diUB-neddylated-CRL2^KLHDC10) and EMDB: EMD-46645 (focused map:UB-KLHDC10-EloB/C). All other data generated for Tables, Figures, and Supplementary Figs. are available in the Source data file. Source data are provided with this paper.

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

## Acknowledgements

We thank the St. Jude Biomolecular X-Ray Crystallography and CryoEM centers for advice, assistance, and support. This research used resources of the Advanced Photon Source, a U.S. Department of Energy (DOE) Office of Science user facility operated for the DOE Office of Science by Argonne National Laboratory under Contract No. DE-AC02-06CH11357. Data were collected at Southeast Regional Collaborative Access Team (SER-CAT) 22-BM beamline at the Advanced Photon Source, Argonne National Laboratory. SER-CAT is supported by its member institutions (https://www3.ser.aps.anl.gov/contact-us#TITLE_SER_CAT_Memberships), equipment grants (S10_RR25528, S10_RR028976 and S10_OD027000) from the National Institutes of Health, and funding from the Georgia Research Alliance. This research also used resources from Berkeley Center for Structural Biology (BCSB). The BCSB is supported by the Howard Hughes Medical Institute, Participating Research Team members, and the National Institutes of Health, National Institute of General Medical Sciences, ALS-ENABLE grant P30 GM124169. The Advanced Light Source is a Department of Energy Office of Science User Facility under Contract No. DE-AC02-05CH11231. The Pilatus detector on beamline 5.0.1 was funded under NIH grant S10OD026941. This research was supported by ALSAC, St. Jude Children's Research Hospital, NIH P30CA021765 to the St. Jude Children's Research Hospital Comprehensive Cancer Center, NCI 5RO1CA247365 (D.C.S. and B.A.S.), NIHR01 GM141409 and NCIR01 CA279255 (N.P. and G.K.), NIH R01GM132129 (J.A.P.), NIH R01AG11085 (J.W.H. and S.J.E.), Howard Hughes Medical Institute (S.J.E.), and Max-Planck-Gesellschaft (B.A.S.). S.J.E. is an Investigator with the Howard Hughes Institute and a member of the Harvard Ludwig Institute.

## Author contributions

D.C.S., S.J.E., G.K., and B.A.S. conceived the project. DCS and MTK generated protein complexes and performed their biochemical characterization. D.C.S. performed biochemical assays, and determined crystal structures. S.C. collected and processed CryoEM data. N.P. and G.K. performed pre-steady state kinetic studies. S.A.M. provided activity based probes for trapping. K.C. and A.N. performed SPR studies. C.L. and J.P. performed and JWH supervised proteomics experiments. D.J.M. collected X-ray crystallography data. D.C.S., G.K., and B.A.S. prepared the manuscript with input from all authors. B.A.S. supervised the project.

## Funding

## Competing interests

D.C.S. and B.A.S. are co-inventors of intellectual property that is unrelated to this work (DCN1 inhibitors licensed to Cinsano). J.W.H. is a founder and consultant for Caraway Therapeutics and is a scientific advisory board member for Lyterian Therapeutics. SJE is a founder of, and holds equity in TScan Therapeutics and Immune ID. S.J.E. is also founder of MAZE Therapeutics, and Mirimus and serves on the scientific advisory board of TSCAN Therapeutics, and MAZE Therapeutics. In accordance with Partners HealthCare's conflict of interest policies, the Partners Office for Interactions with Industry has reviewed SJE's financial interest in TSCAN and determined that it creates no significant risk to the welfare of participants in this study or to the integrity of this research. B.A.S. is a member of the scientific advisory boards of Biotheryx and Proxygen. The remaining authors declare no competing interests.
