## [Transparent Peer Review file · Nature Communications]

Structural basis for C-degron selectivity across KLHDCX-family E3 ubiquitin ligases

Corresponding Author: Professor Brenda Schulman

Version 0:

Reviewer comments:

Reviewer #1

(Remarks to the Author)

Review of NCOMMS-24-52447

A seminal 2018 publication in Cell by Koren et al. revealed several new principles of substrate and degron recognition by components of the ubiquitin-proteasome system (UPS) and inspired numerous questions. Among the most alluring of those questions has been addressed by the current manuscript from Scott et al. The manuscript provides beautiful structural and biochemical detail as to how a family of cullin RING ligases, known as KLHDCx, recognize distinct substrate proteins despite these substrates having similar C-end degrons terminating in a critical glycine residue. The manuscript is well organized and the results will be of great interest to the UPS field and more generally those who study proteins containing beta-propeller domains. I feel this manuscript will be ready for publication in Nature Communications with only minor edits and additions. I have only a few comments for the authors. Very nice!

- 1) I suggest a bit more introduction and discussion of the potential cellular and biological significance of UBL ubiquitylation. I know this was somewhat covered in the 2023 Scott et al. Molecular Cell paper, but readers who do not read that and other papers may be confused as to the use of these proteins as model substrates for KLHDCx E3s. Do CRL2-KLHDC3 or CRL2-KLHDC10 create cellular UB chains, perhaps constantly in flux because of cellular DUB activity? I do not suggest experiments, just a bit of discussion.
- 2) For Figure 1 legend, I suggest adding the detail that the monomeric versions of KLHDCx were used.
- 3) Figure 2e: I suppose the electrostatic representation is standardized enough to not need explanation but consider a key or brief explanation in legend—for quick reading by non-structural biologists.
- 4) Figure 3b legend: disruption of tetramer is inferred from the experiment and data (and almost certainly correct), but the data shown (gels for UB) do not show this directly. I suggest re-wording.
- 5) I presumed that “SELK” in Figures 1 and 6 refers to a peptide (smaller than the entire protein) model substrate of KLHDCx, but I was not sure. I suggest clarifying and adding a bit more detail to the methods on this substrate (labeled version).
- 6) There are several spelling errors throughout (e.g., first paragraph of discussion; figure legends). I expect most will be corrected by editors, but sometimes errors in figures and graphs are missed. See Y-axis of graphs in Figure 1.

Reviewer #2

(Remarks to the Author)

This manuscript by Scott et al. investigated the C-degron recognition mechanism of several KLHDC family members. Using primarily biochemical and structural biology approaches, the authors elucidated the general principles underlying degron recognition and highlight important differences among family members. The manuscript is clearly written, and the structures, determined using X-ray crystallography and single-particle cryo-EM, are of high quality. The interpretations of the experimental data are robust, and the results demonstrate the specificity and plasticity of degron recognition, which may be applicable to other ubiquitin systems.

This reviewer has no major concerns. However, there are several points the authors should consider clarifying in their revision:

1. Does the R/F-S-R motif appear in all six blades of the KLHDC family? In other words, given a substrate other than ubiquitin, do the authors expect the C-degrons to always interact with the same blade as observed in the ubiquitin cases? More broadly, how general is the model proposed in Fig 7?
2. Fig 2j presented several ubiquitin mutants that affected ubiquitylation, but it is unclear where these residues are located without mapping to the crystal structure. Were these residues chosen randomly? Do they all affect KLHDC3 binding? The same question applies to Fig 4i.
3. It is surprising that UBTCAP showed a much higher K_d for KLHDC3. The current explanation provided by the authors, i.e., "residues upstream of TCAP's C-terminal RG motif interact unfavorably with KLHDC3", requires further clarification.
4. In Fig 6a it is unclear which parts are aligned, only the acceptor UB and the UB in the KLHDC3 complex? It would be helpful to include the superimposition in the context of the entire complex (8PQL). Also, do the authors mean that the structural alignment suggested a possible "faster priming" when ubiquitin is the substrate, which was later supported by the results? This rationale is somewhat unclear in the text.
5. Did the authors attempt to determine the structure of KLHDC10 in complex with UB(G75W)? This would further support their conclusions. Given the complexity of cryoEM for such a large complex, it is understandable if they chose not to pursue this.

Minor points:

1. Fig 7 should be cited in the Discussion section, as it is not currently referenced in the manuscript.
2. Fig 2b and 2c appear too "foggy".
3. In Fig 4a, please include the actual experimental map alongside the model.
4. It would be beneficial to include a figure highlighting the hydrophobic pocket in the KLHDC10 structure consisting of L127, A155, Y110, and F176.
5. The resolution of the crystal structures could potentially be extended based on CC1/2 and I/σ , particularly for 9D11.
6. Space group: C2221 is preferred.
7. The angles in the unit cell dimensions should be represented by Greek letters (which were automatically corrected to a, b, g in the manuscript).

General response to reviewers:

We are pleased by the enthusiastic responses from the Reviewers! It is gratifying to receive such supportive comments regarding our work, thank you! We are also grateful to the Reviewers for detailed reading revealing several spelling errors and mistakes. We tried to address all suggestions by revising the text and figures.

Reviewer #1 (Remarks to the Author):

Review of NCOMMS-24-52447

A seminal 2018 publication in Cell by Koren et al. revealed several new principles of substrate and degron recognition by components of the ubiquitin-proteasome system (UPS) and inspired numerous questions. Among the most alluring of those questions has been addressed by the current manuscript from Scott et al. The manuscript provides beautiful structural and biochemical detail as to how a family of cullin RING ligases, known as KLHDCx, recognize distinct substrate proteins despite these substrates having similar C-end degrons terminating in a critical glycine residue. The manuscript is well organized and the results will be of great interest to the UPS field and more generally those who study proteins containing beta-propeller domains. I feel this manuscript will be ready for publication in Nature Communications with only minor edits and additions. I have only a few comments for the authors. Very nice!

We thank the Reviewer for such kind comments and enthusiasm for our work!

1) I suggest a bit more introduction and discussion of the potential cellular and biological significance of UBL ubiquitylation. I know this was somewhat covered in the 2023 Scott et al. Molecular Cell paper, but readers who do not read that and other papers may be confused as to the use of these proteins as model substrates for KLHDCx E3s. Do CRL2-KLHDC3 or CRL2-KLHDC10 create cellular UB chains, perhaps constantly in flux because of cellular DUB activity? I do not suggest experiments, just a bit of discussion.

We have added additional an additional paragraph in the Introduction to more clearly define UBL ubiquitylation by KLHDCX family members. In addition, we have added a new paragraph in the Discussion to speculate on the potential for KLHDCX family members (specifically KLHDC3 and unanchored K48 linked ubiquitin chains) to promote UBL ubiquitylation in cells and possible biological implications.

2) For Figure 1 legend, I suggest adding the detail that the monomeric versions of KLHDCx were used.

We now indicate in the figure legends where monomeric KLHDC3 was utilized in experiments.

3) Figure 2e: I suppose the electrostatic representation is standardized enough to not need explanation but consider a key or brief explanation in legend—for quick reading by

non-structural biologists.

We added brief explanations of the electrostatic representations in the figure legends.

4) Figure 3b legend: disruption of tetramer is inferred from the experiment and data (and almost certainly correct), but the data shown (gels for UB) do not show this directly. I suggest re-wording.

We reworded this statement accordingly.

5) I presumed that “SELK” in Figures 1 and 6 refers to a peptide (smaller than the entire protein) model substrate of KLHDCx, but I was not sure. I suggest clarifying and adding a bit more detail to the methods on this substrate (labeled version).

The SELK used in biochemical experiments is a fragment from residue 43 to the C-terminus, representing the complete cytoplasmic domain of SELK. We have added additional details to the methods section to indicate this and describe its expression and purification and that we refer to this as "SELK" in the text and figures.

6) There are several spelling errors throughout (e.g., first paragraph of discussion; figure legends). I expect most will be corrected by editors, but sometimes errors in figures and graphs are missed. See Y-axis of graphs in Figure 1.

We thank the reviewer for reading the manuscript in detail and pointing out the spelling errors. We have gone through the text and figures to correct mistakes.

Reviewer #2 (Remarks to the Author):

This manuscript by Scott et al. investigated the C-degron recognition mechanism of several KLHDC family members. Using primarily biochemical and structural biology approaches, the authors elucidated the general principles underlying degron recognition and highlight important differences among family members. The manuscript is clearly written, and the structures, determined using X-ray crystallography and single-particle cryo-EM, are of high quality. The interpretations of the experimental data are robust, and the results demonstrate the specificity and plasticity of degron recognition, which may be applicable to other ubiquitin systems.

We are delighted by the Reviewer's enthusiasm and support of our work.

This reviewer has no major concerns. However, there are several points the authors should consider clarifying in their revision:

1. Does the R/F-S-R motif appear in all six blades of the KLHDC family? In other words, given a substrate other than ubiquitin, do the authors expect the C-degrons to always

interact with the same blade as observed in the ubiquitin cases? More broadly, how general is the model proposed in Fig 7?

The R/F-S-R motif is unique to the blades that capture the degron C-terminus. As such, we anticipate that each KLHDCX family member would anchor various C-degrons in the same way. Indeed, there are 6 structures of the isolated KLHDC2 propeller bound to C-degrons or C-degron-like sequences and they all show the C-terminus anchored to the R-S-R motif. To showcase the unique locations of the motifs, we now include a new Supplementary Figure (Fig S9) which shows sequence alignments over the regions containing the R/F-S-R motifs which clarifies that the sequences are not conserved in the corresponding regions of KLHDCX family members whose R/F-S-R motifs are in different blades.

2. Fig 2j presented several ubiquitin mutants that affected ubiquitylation, but it is unclear where these residues are located without mapping to the crystal structure. Were these residues chosen randomly? Do they all affect KLHDC3 binding? The same question applies to Fig 4i.

The structural panels in Fig 2i show the globular domain residues of UB tested, and their relative proximity to potential interacting residues from KLHDC3. The UB mutations made and tested were based on this structural insight. For KLHDC10, flexibility of UB in the complex precluded unambiguous placement of the UB globular domain within the cryoEM density. However, because the organization was reminiscent of that observed for KLHDC3, we tested the same panel of UB mutants surveyed with KLHDC3. These data are consistent with the globular domain of UB also playing a role in the recognition and ubiquitylation of UB by KLHDC10.

3. It is surprising that UBTCAP showed a much higher K_d for KLHDC3. The current explanation provided by the authors, i.e., “residues upstream of TCAP’s C-terminal RG motif interact unfavorably with KLHDC3”, requires further clarification.

We thank the reviewer for spotting this and apologize for this error in description. The sentence should have read “residues upstream of TCAP’s C-terminal RG motif interact FAVORABLY with KLHDC3”. We have corrected this in the text.

4. In Fig 6a it is unclear which parts are aligned, only the acceptor UB and the UB in the KLHDC3 complex? It would be helpful to include the superimposition in the context of the entire complex (8PQL). Also, do the authors mean that the structural alignment suggested a possible “faster priming” when ubiquitin is the substrate, which was later supported by the results? This rationale is somewhat unclear in the text.

We have added additional information in the figure legend to indicate how the structures were superimposed. Since UBE2R2 has been visualized in multiple conformations for modifying CUL2 substrates we are hesitant to superimpose the structure in other ways, as we anticipate flexibility of the E3 can allow UBE2R2 to

sample multiple conformations. The suggestion of faster priming derives from the unique case for KLHDCX family members in that ubiquitin itself is the substrate for priming. The structural superimposition is included as a reference to readers indicating that the same surfaces used for polyubiquitylation by UBE2R2 with other CRL2 E3s is the same surface utilized for priming of a UB bound KLHDC3 or KLHDC10 E3.

5. Did the authors attempt to determine the structure of KLHDC10 in complex with UB(G75W)? This would further support their conclusions. Given the complexity of cryoEM for such a large complex, it is understandable if they chose not to pursue this.

As noted by the reviewer, attaining high resolution insights into the recognition by KLHDC10 required very challenging sample preparation. As such, we did not determine the structure of a G75W ubiquitin.

Minor points:

1. Fig 7 should be cited in the Discussion section, as it is not currently referenced in the manuscript.

Thank you for pointing out this omission. We now reference Fig. 7 in the discussion section.

2. Fig 2b and 2c appear too “foggy”.

We have adjusted the depth cue and transparency levels to make these structure panels appear less “foggy”.

3. In Fig 4a, please include the actual experimental map alongside the model.

We now include a panel in Fig 4a showing the experimental map.

4. It would be beneficial to include a figure highlighting the hydrophobic pocket in the KLHDC10 structure consisting of L127, A155, Y110, and F176.

We have changed the surface representations in Fig. 5b, c to electrostatic representations to show the hydrophobic pocket formed by L127, A155, Y110, and F176.

5. The resolution of the crystal structures could potentially be extended based on CC1/2 and $1/\sigma$, particularly for 9D1I.

Unfortunately, during the synchrotron data collection the detector was not placed to capture the higher resolution reflections.

6. Space group: C2221 is preferred.

Yes, the structures were determined in C2221.

7. The angles in the unit cell dimensions should be represented by Greek letters (which were automatically corrected to a, b, g in the manuscript).

We have corrected the mistake in the crystallographic table.